# Identifying the multiple drivers of cactus diversification

Jamie B. Thompson [1,2] ✉, Tania Hernández-Hernández[3], Georgia Keeling[2], Marilyn Vásquez-Cruz [4] & Nicholas K. Priest[2]

Our understanding of the complexity of forces at play in the rise of major angiosperm lineages remains incomplete. The diversity and heterogeneous distribution of most angiosperm lineages is so extraordinary that it confounds our ability to identify simple drivers of diversification. Using machine learning in combination with phylogenetic modelling, we show that five separate abiotic and biotic variables significantly contribute to the diversification of Cactaceae. We reconstruct a comprehensive phylogeny, build a dataset of 39 abiotic and biotic variables, and predict the variables of central importance, while accounting for potential interactions between those variables. We use state-dependent diversification models to confirm that five abiotic and biotic variables shape diversification in the cactus family. Of highest importance are diurnal air temperature range, soil sand content and plant size, with lesser importance identified in isothermality and geographic range size. Interestingly, each of the estimated optimal conditions for abiotic variables were intermediate, indicating that cactus diversification is promoted by moderate, not extreme, climates. Our results reveal the potential primary drivers of cactus diversification, and the need to account for the complexity underlying the evolution of angiosperm lineages.

The angiosperm family Cactaceae is an iconic component of ecosystems spanning the Americas[1–3]. Nearly all cacti exhibit the succulent life form which enables survival in the face of water scarcity, through adaptations including succulent stems, reduction, modification or loss of leaves, and crassulacean acid metabolism (CAM) photosynthesis[4–6]. Although cacti are found across diverse ecosystems including wet tropical forests and colder regions[7], their richness is highest in arid and semi-arid regions[2,4]. This has implicated aridification as a central driver of diversification in cacti, which have some of the fastest diversification rates across plants despite long generation times[1,3,8], and succulents generally[4,5]. However, aridification struggles to explain dramatic within-family phylogenetic imbalances. Major cactus radiations occurred much more recently than the onset of aridity at their locales and are primarily associated with the evolution of novel growth forms and reproductive strategies[2,3]. Further forces shape diversification

rates of specific cactus groups including temperature[9]. Furthermore, recent research suggests that speciation rates in cacti are fastest in semi-arid to humid regions[10]. Thus, drivers other than increased aridification must play a key role in shaping rate heterogeneity within cacti[2]. We need to develop our understanding of the forces shaping cactus diversity, one of the most endangered of any major taxonomic lineage[11].

The extraordinary ecomorphological diversity of cacti implicates biotic drivers of diversification, with several identified. Growth forms range from small button-like species in genera including *Epithelantha*, to massive columnars such as *Pachycereus*[12], and pollination syndrome varies across bees, moths, birds, and bats[2,13]. Diversification rates are fastest in species with larger growth forms and derived pollination syndromes (bat, bird and moth pollination)[2,14]. However, a range of other variables are linked to cactus biodiversity patterns, including

---

[1]School of Biological Sciences, University of Reading, Whiteknights, Reading, Berkshire, UK. [2]The Milner Centre for Evolution, Department of Life Sciences, University of Bath, Bath, United Kingdom. [3]Department of Research, Conservation and Collections, Desert Botanical Garden, Phoenix, AZ, USA. [4]Instituto Tecnológico Superior de Irapuato, Tecnológico Nacional de México, Irapuato, Guanajuato, México. ✉e-mail: j.b.thompson@reading.ac.uk

distribution[2,7,9], elevation[15], temperature[16], chromosome number[17], edaphic properties[18] and climatic variables[19]. Due to the complex ecological interplay among these variables, important drivers can be obscured, and we do not fully understand which are crucial for cactus diversification. Furthermore, we do not fully understand the impacts of drivers known to shape other succulent lineages (e.g., topographic complexity[20]), or angiosperms as a whole (e.g., spatial distribution[21]).

Macroevolutionary research typically investigates a small number of drivers, neglecting the complexity underlying diversification. This is exemplified in cacti by the dependent evolution of growth form and derived pollination syndromes[2]. Establishing whether diversification rate is truly dependent on one, or both, is difficult. This problem is amplified when considering dozens of potential drivers. Although recent advances in hidden-state models provide a more sophisticated framework for confirming drivers[22], these models are computationally expensive and cannot integrate dozens of variables. Additionally, these cannot handle continuous variables such as aridity index, or model non-linear relationships. A complete understanding of the complexity underpinning cactus diversification requires thorough investigation of widely sampled drivers, whilst accounting for their interactions. One method of doing so is implementing machine learning methods to efficiently filter important variables from a pool while considering interactions[23]. This has revealed key drivers of reef fish diversification[24] and has the potential to explain the complexity of cactus diversification.

Here, we apply machine learning and phylogenetic methods to help explain the mystery of cactus diversification. We reconstruct a phylogeny containing over a thousand cactus species and assemble a dataset of 39 potential predictors. Many of these are hypothesised to shape diversification or diversity patterns of cacti, while several have impacts across succulents and angiosperms generally. We ranked the relative importance of drivers of diversification with XGBoost[23], while assessing the impact of potential interactions between drivers by comparing interaction models with an equivalent "stump" model (maximum interaction depth of one). We then used state-dependent speciation and extinction (SSE) models to confirm the impact of significant drivers. Our results indicate that cactus diversification is shaped primarily by diurnal air temperature range, soil sand content and plant size (height or length). Minor impacts are also conferred by isothermality and geographic range size. Significantly, hypotheses of growth form variation, pollinator divergence and aridity are recovered as not primarily important, when accounting for complex interactions, and sampling drivers broadly. Our results reinforce the complexity of biological diversification, as well as the difficulty of identifying key factors shaping the rise of cacti.

## Results

### Heterogeneous diversification of Cactaceae

We reconstructed a Maximum Likelihood phylogeny of 1063 cacti and six Caryophyllales outgroups, using a supermatrix which sampled 15 plastid and three nuclear loci (16.47 kb, 6,020 parsimony-informative sites and 77.5% missing data). Our phylogeny is moderately well-supported, with 32.3% of nodes supported by >70% BS and 11.9% by >90% BS. Many of the shallower nodes are relatively weakly supported, which is a common finding in phylogenetic analyses of cacti[13,25–27]. Furthermore, topology and estimated divergences are broadly congruent with previous hypotheses[1,2]. We estimate the stem age of Cactaceae as ~48.51 Mya, and the crown age as ~37.24 Mya (splitting of *Leuenbergeria* from the remainder). Stem ages of major lineages are similar, with *Pereskia* at 36.99 Mya, Maihuenioideae and Opuntioideae at a near-simultaneous 36.86 Mya and Cactoideae at 35.76 Mya. Crown ages of *Leuenbergeria*, *Pereskia*, Maihuenioideae, Opuntioideae and Cactoideae are 19.65, 34.94, 4.03, 17.13 and 35.76 Mya, respectively.

We find diversification rate is remarkably varied (Fig. 1), recovering strong support for heterogeneity using BAMM[28] (Bayes Factor

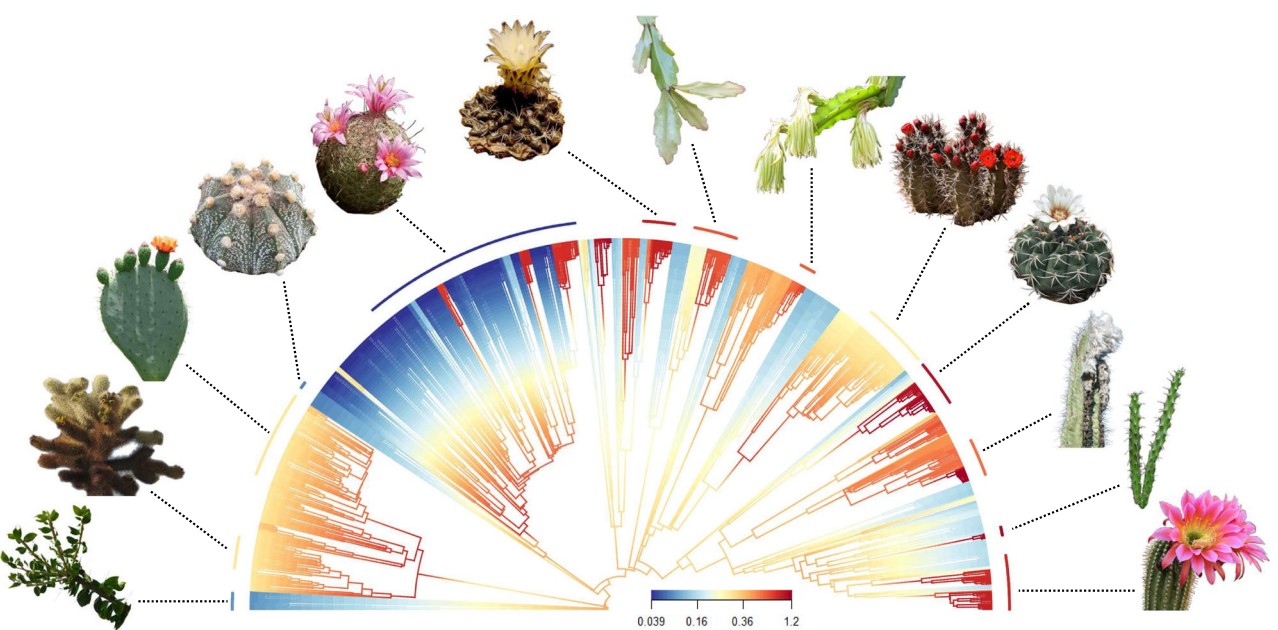

0.039  0.16  0.36  1.2

**Fig. 1 | Remarkable diversification rate heterogeneity across Cactaceae.** Branches are coloured by speciation rates estimated with BAMM28 and vary 32-fold. Arc segments of median speciation rate for thirteen morphologically varied cactus genera are indicated. Cactus images are used under Creative Commons with modifications allowed. From left to right: images 1, 3, 8, 11, 12, and 13 used photos taken by Amante Darmanin, Forest & Kim Starr, John Tann, Renee Grayson, and Wendy Cutler, which are licensed under a Creative Commons Attribution 2.0 License (https://creativecommons.org/licenses/by/2.0/). Image 2 used a photo marked as being in the Public Domain (https://creativecommons.org/publicdomain/mark/1.0/). Images 4 and 10 used photos taken by Leonora Enking and Lyubo Gadzhev, which are licensed under a Creative Commons Attribution-ShareAlike 2.0 License (https://creativecommons.org/licenses/by-sa/2.0/). Images 5, 7 and 9 used photos marked as being in the Public Domain using the CC0 1.0 Universal Public Domain Dedication (https://creativecommons.org/publicdomain/zero/1.0/). Image 6 used a photo taken by Christer Johansson, which is licensed under a Creative Commons Attribution 3.0 Unported License (https://creativecommons.org/licenses/by/3.0/deed.en).

4,096). The best-supported rate shift configuration has 20 shifts (Supplementary Fig. 1), and tip-speciation varies 32-fold, from <0.04 in some *Mammillaria* to >1.2 in *Gymnocalycium* and *Pilosocereus*. Despite the richness of Opuntioideae, only a single rate shift is recovered in the basal branch. Multiple shifts were recovered across Cactoideae, including several within the *Mammillaria* complex, and shifts in groups Cereeae, Trichocereeae, Cacteae, Hylocereeae, and Browningieae. Although not relevant to the focus of the current study, we also find that species richness and tip-speciation rates vary across the Americas, confirming that cacti are a good system for identifying drivers of diversification across diverse ecological contexts (Supplementary Fig. 2).

## Multiple drivers of cactus diversification

Our dataset of potential drivers consists of 39 explanatory variables for up to 1063 ingroup species, with 21.59% missing data and a mean of ~834 entries per variable, ranging from 374 (chromosome count) to 1063 (growth form). Thirty-five variables were estimated from global models of climatic, topographic and edaphic variables, using GBIF data which, after curation, comprised 9485 coordinates for 850 species. The full dataset is available in Supplementary Data (https://github.com/jamie-thompson/cactaceae).

We find that five variables are significant predictors of tip-speciation rates when accounting for complex interactions with XGBoost, of which four are abiotic and one biotic (Fig. 2a). The primary driver is mean diurnal air temperature range (bio2), and weaker predictive power is found for soil sand content, plant size (height or length), isothermality (bio3), and geographic range size. The full XGBoost model had very high prediction accuracy (mean bias = 0.034)

and moderate precision (mean $R^2 = 0.21$). No significant correlations are found among the significant variables, indicating successful selection of important features (Supplementary Fig. 5). In the XGBoost model not accounting for complex interactions (the "stump" model), predictive power was reduced (mean $R^2 = 0.15$; Fig. 2b), and the rank order of variables shifted. Three variables (biome, precipitation seasonality and chromosome count) were identified as significant predictors in the simple model, but are non-significant in the complex model. Isothermality and geographic range size were not significant in the simple model but are significant in the complex model (Fig. 2). Though some changes are small, the rise in importance of isothermality and geographic range size is notable and shows that complexity can obscure our ability to detect the central drivers of diversification (Fig. 2).

XGBoost accounts for outliers; nevertheless, we ran additional analyses to test how the findings are affected by unusually high estimated speciation rates. In analyses excluding species with tip-speciation rates above 0.65, we find that strong predictive power is maintained for soil sand content and plant size, but is reduced in the most powerful predictive parameter, diurnal air temperature range, while there is minor variation in the weakest significant variables (Supplementary Data, https://github.com/jamie-thompson/cactaceae).

## Relationships between drivers and diversification

Each of the five significant predictors of tip-speciation rates are confirmed to relate to cactus diversification, with QuaSSE[29] models (Table 1, Fig. 3). The best-fitting relationship with speciation rates is modal with drift, for all five variables (Table 1, Fig. 3). Interestingly, each of the computed parameter estimates for the abiotic variables

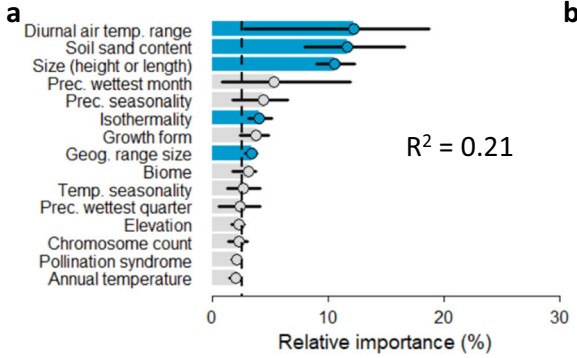
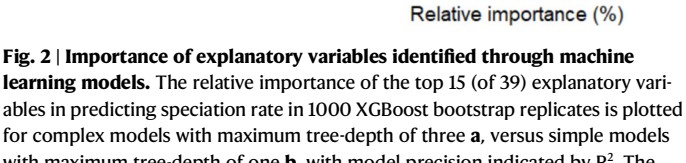
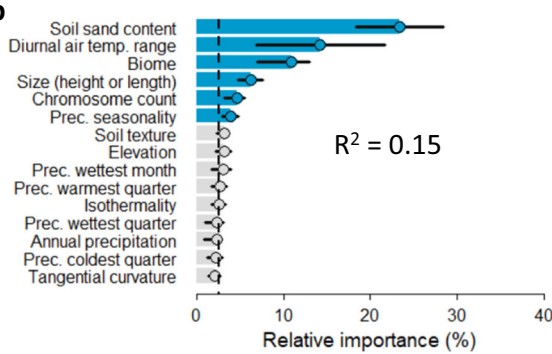

**Fig. 2 | Importance of explanatory variables identified through machine learning models.** The relative importance of the top 15 (of 39) explanatory variables in predicting speciation rate in 1000 XGBoost bootstrap replicates is plotted for complex models with maximum tree-depth of three **a**, versus simple models with maximum tree-depth of one **b**, with model precision indicated by $R^2$. The vertical dashed line indicates the threshold of predicting speciation rate by chance expectation alone. Upper and lower importance quantiles (25% and 75%) estimated from 1000 model bootstraps are indicated with black horizontal bars. When interactions are accounted for, the relative importance of several variables shifts, and the $R^2$ is improved.

**Table 1 | Model comparisons of QuaSSE analyses of continuous drivers with significant predictive ability in the XGBoost model, with AIC scores of the best-fitting and null models**

| Variable and sample size | Best-fitting model | ΔAIC of best fitting model versus constant model (null) | Midpoint value (back-transformed to the original scale if they were log-transformed), and drift parameter |
|---|---|---|---|
| Mean diurnal air temperature range ($n = 850$) | Modal with drift | 233.98 | 10.20 °C, 0.024 |
| Soil sand content ($n = 850$) | Modal with drift | 65.53 | 48.38%, 0.00045 |
| Plant size (height or length) ($n = 750$) | Modal with drift | 76.72 | 45.74 cm, −0.060 |
| Isothermality ($n = 850$) | Modal with drift | 237.10 | 0.55, 0.0012 |
| Geographic range size ($n = 850$) | Modal with drift | 307.86 | 0.23 AOO, −0.45 |

Delta AIC is >4 for all best-fitting models versus second best-fitting models with $p < 0.0001$ for each of the best-fitting models.

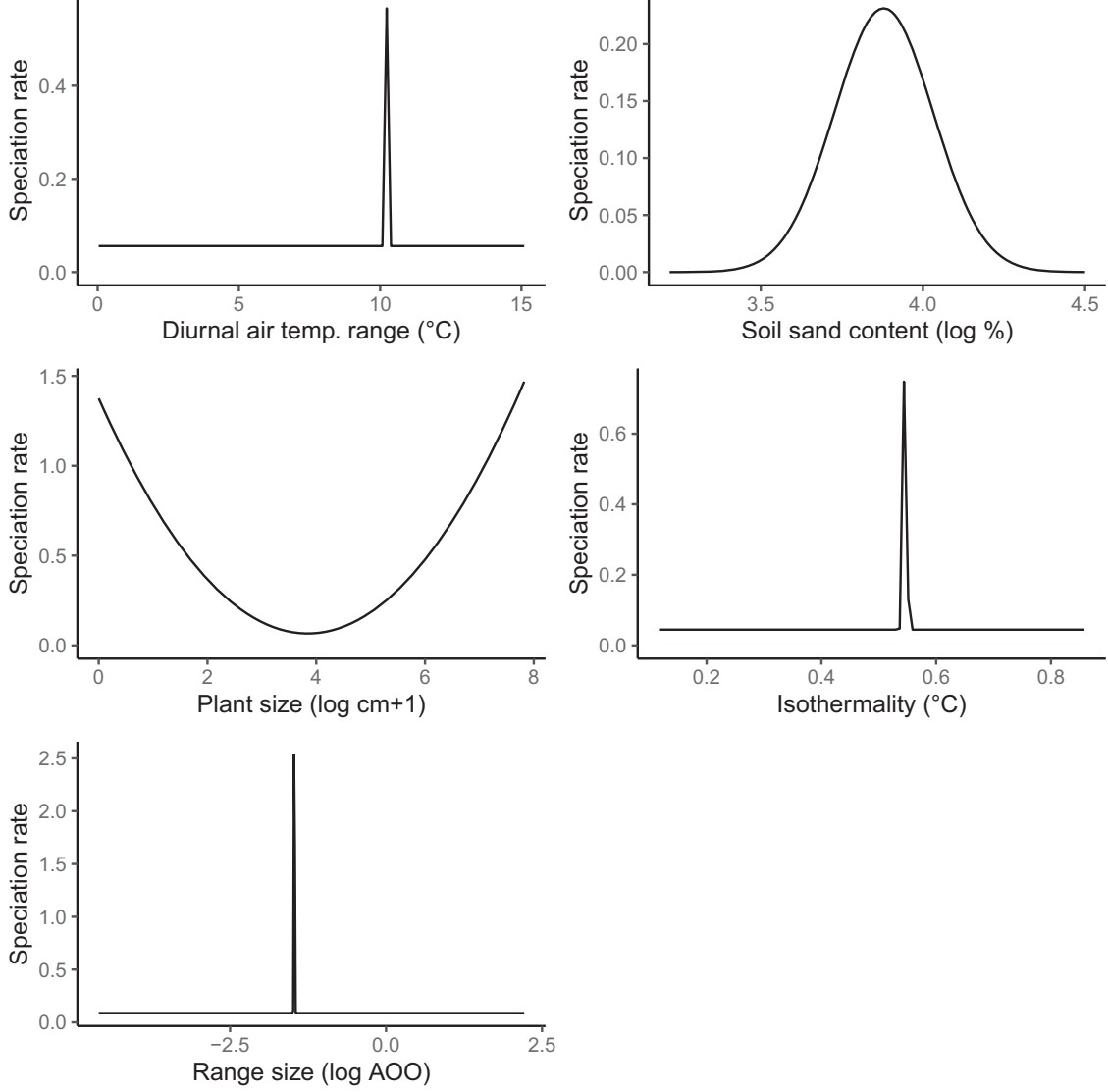

**Fig. 3 | Best-fitting relationships between continuous variables inferred as significant by XGBoost and speciation rate, as estimated by QuaSSE.** Variables are in order of ranked importance according to the full XGBoost model. It is important to consider that QuaSSE does not provide the exact relationship between variables and speciation rates, only the general trend[29]. The reported model fits are those best-supported by the data. The cases with narrow modes are likely shaped by the inability of QuaSSE to account for hidden states and confounding correlations between variables. We present these as hypotheses to inform future research.

were intermediate, indicating that the speciation rates in cacti are generally accelerated by moderate, not extreme, climatic variability. The modal models were strongly supported over the second best-fitting models for every variable (ΔAIC > 4).

Some estimates of variances in modal models are small and we do not report them (Fig. 3). Though not pertaining to the focus of our study, for completeness, we report estimates of drift, the directional tendency of the variable across evolutionary time. It is important to note that QuaSSE is limited to estimating general trends, and not the true relationship between variables and diversification[29]. Predicted speciation rates by XGBoost can show higher resolution, but they show different patterns to QuaSSE, because they do not account for the effects of shared ancestry (Supplementary Fig. 6).

## Discussion

Multiple forces have been proposed to shape diversification of the Cactaceae, but it is difficult to extricate key drivers. With a large phylogeny and extensive dataset of abiotic and biotic variables, we characterise the complexity of cactus diversification with machine learning and explore their relationships with phylogenetic SSE models. Our analyses support the role of five primary drivers of diversification, including previously under-investigated variables, such as geographic range size. We provide further support for previously hypothesised forces including plant size, but find little support for others including growth form, pollination syndrome and aridity. Our results reinforce the complexity of macroevolution, but also the difficulty of predicting diversification rates. The XGBoost model identifies significant unexplained variance in cactus diversification. Furthermore, these results have bearing on the long-term impacts of global climate change on the biodiversity of cacti, an area of active research[7].

Though complexity in cactus diversification was previously explored by associating radiations with two coevolving traits[2], a comprehensive understanding of forces is unknown. Our XGBoost models emphasise the need to unravel these interactions, with lower predictive power for the simple model than the complex model. Failure to account for interactions resulted in incorrect inference of relative importances, exemplified by the reduction of importance of

diurnal air temperature range, and the elevation of biome. Our results contribute to recent research exploring complexity in macroevolution[22,24] and call for a more thorough examination across the Tree of Life. Given the low explanatory power of individual drivers explained XGBoost, and the discrepancy between XGBoost predictions and phylogenetically controlled QuaSSE results (Supplementary Fig. 6), which provide only a general trend, further research is needed to clarify exact relationships between drivers and cactus diversification. With this in mind, we wish to present discussion on the five drivers simply as hypotheses to shape future research.

Adaptations to temperature variation may help explain our identification of a modal relationship between cactus speciation and diurnal air temperature range, with a relatively low optimum of 10.20 °C, as estimated by QuaSSE. Cacti are commonly thought of as the hardiest plants, however, they are finely-tuned to their environments[30]. Although extreme deviations in diurnal temperature are negatively associated with population survival[7], the model fits show that speciation rate is highest at 10.2 °C, an intermediate value of diurnal temperature range. Cacti have adapted to withstand diurnal temperature ranges by minimising water loss. Such adaptations include species of *Mammillaria* and *Carnegeia*, which alter their stem diameter to mitigate the impact of daily temperature fluctuations[31]. Notwithstanding, temperature extremes pose distributional limits in cacti[32], and across all plants. Many plants, including cacti, struggle to withstand high temperatures[33–35], with photosynthetic performance of cacti reducing upon 2 °C of heating[36]. Increased temperatures affect germination[37], and population or range sizes[7,38]. Conversely, most succulents struggle to avoid freezing damage at low temperatures[6], strongly constraining some species distributions, which may act as a selective pressure shaping species diversity[39,40]. Therefore, physiological limits encountered in environments with high diurnal air temperature variation likely explain the finding of optimal speciation rates in environments with relatively low diurnal temperature variations, through survivorship variation. It is unclear how this variation explicitly links to differential rates of reproductive isolation though. Further studies are necessary to establish the relationship between morphological adaptations, speciation, and diurnal temperature variation.

Soil is an important determinant of plant biodiversity[41,42], including cacti[19], and has been related to diversification[43]. We found that soil sand content is the third best predictor of cactus speciation rates (Fig. 3), with fastest speciation in medium-sandy soils, as estimated by QuaSSE. Edaphic variation determines which species can grow and survive in particular regions, and different species have different requirements[44]. Particular conditions may promote survival, reproduction and population growth in well-adapted species, which leads to reproductive isolation through ecological specialisation, with non-optimal conditions being detrimental to macroevolutionary success[45]. Sand content shapes diversity patterns of the cactus genus *Neobuxbaumia*[18], and species inhabiting the Caatinga[46]. Sand content is a relevant property for cacti, associated with poor water retention and fertility[47,48], which provides selective pressure for shallow roots in many cacti. Soils with high levels of sand may be too poor at retaining water even for arid-adapted cacti, which reduces the opportunity for population growth and reproductive isolation. Furthermore, soils with high sand content create edaphically dry areas, which may have extreme precipitation and temperature patterns, predicted to be detrimental for cactus populations[7]. In contrast, where soils are less sandy, plants without adaptations to sandy soil are more likely to thrive, providing competition to cacti.

Plant size is the third most powerful predictor of speciation (Fig. 2). Plant size has many physiological and ecological consequences[49], allometrically scaling with numerous life history traits[50], including those with powerful impacts on speciation rates across angiosperms[51]. An inverted modal relationship between cactus size and speciation rate is best supported by QuaSSE. QuaSSE cannot provide the exact relationship between plant size and diversification rate, though it can provide the general trend and test significance of a variable[29], and here we limit description to broad-scale patterns. QuaSSE estimated fastest rates in the smallest and largest cacti, and reduced rates in intermediate sizes (Fig. 3). Here, the smallest cacti tend to be species scored as 0 (globose caespitose, globose solitary and barrel), while the largest tend to be species scored as 1 (arborescent, shrubby and treelike). Although this growth form binarisation provides limited fine-scale resolution, it has been shown to be useful given the difficulty in scoring growth forms in cacti[2,13]. Furthermore, growth form strongly shapes cactus size variation (Supplementary Fig. 3).

Relationships between organismal size and speciation rate are frequently documented in lineages across the Tree of Life[52,53], but the nature of trends can vary[54]. It is typically thought that smaller species speciate more rapidly due to faster mutation rates and generation times, reduced gene flow, and higher selection on more niche axes, leading to reproductive isolation[50]. While we do recover this classic pattern, we also find that speciation rates of larger cacti are elevated, with lowest rates found in intermediate sizes (Fig. 3). This pattern has been documented previously in subfamily Cactoideae and is explained through a coevolution with pollination syndrome[2]. Pollinator divergence can compensate for reproductive difficulties conferred by arid biomes, in which mate-finding Allee effects are amplified due to low population densities[55]. Derived pollination syndromes can provide barriers to gene flow[56], facilitating reproductive isolation[57]. Furthermore, bat and bird pollinators deposit a larger amount of pollen, and disperse over longer distances than ants or bees (the ancestral state)[58]. Variation in dispersal distance can influence reproductive isolation because it determines gene flow within and between populations[59,60]. Why intermediately sized cacti speciate more slowly could result from growth form variation, which strongly shapes size variation (Supplementary Fig. 2), and is associated with different evolutionary strategies, and reproductive isolation mechanisms.

Previous research in climate-driven diversification has focused on impacts of historic climatic change[27,61–63] and broad-scale spatial variation[10,64,65], but less is understood about the impacts of localised temperature variability. We find that isothermality, the ratio of diurnal and annual temperature variation, is a significant predictor of cactus speciation rate. As with diurnal air temperature range, speciation is fastest at intermediate isothermality values (~0.55), as estimated by QuaSSE. This optimum represents environmental conditions where diurnal air temperature fluctuations are ~55% the magnitude of annual fluctuations. In these conditions, seasonality is more powerful than daily variation, but daily temperature does fluctuate. These conditions could plausibly accelerate speciation through seasonal variation in temperature driving niche evolution, but with relatively unexceptional daily fluctuations reducing immediate stressors to cacti. However, isothermality as a driver of evolution is understudied compared to other climatic variables such as precipitation and temperature.

High isothermality values indicate environments of powerful diurnal temperature variability relative to annual. One region of Earth characterised by high isothermality is the Tropics, which have little seasonal variation but strong diurnal fluctuations[66]. Outside of the Tropics, where annual variation is pronounced, species richness of cacti is highest[4,7] (Supplementary Fig. 2). Our results, which indicate that cactus speciation is fastest when climatic conditions are unexceptional, support recent research which found that cacti are more vulnerable to climatic variation than commonly thought[7]. It is becoming clear that cacti, typically thought of as among the hardiest of all plants, are not especially robust to extremes of climate, which has significant consequences for the conservation of succulent floras.

Geographic range size is a complex biotic variable influenced by numerous ecological and evolutionary factors[67]. Though range size determines family-level richness across angiosperms, with widespread

families diversifying more quickly[68], the impact at lower taxonomic levels is poorly understood. We find that range size has a weak but significant impact on cactus speciation. We wish to promote caution in interpretation of these results, since geographic range size is the weakest of the significant predictors by XGBoost (Fig. 2), is the most sensitive variable to sampling issues pervading GBIF, and is difficult to estimate for cacti especially, as detailed in the methods. Furthermore, the QuaSSE fit has the tightest variance of all, which suggests it is shaped primarily by a few rapidly-speciating species instead of a general trend. We discuss this speculatively, to inform future research, which we urge is critical, as others have before[7,11]. We find, using QuaSSE that speciation is fastest in species with ~0.23 AOO (Fig. 3). This is at the low to medium end of range sizes in cacti. It is characteristic of species including in *Opuntia tehuacana*, *Stenocereus montanus* and *Facheiroa squamosa*, which are found across individual regions within countries: Southern Mexico, Northern Mexico, and Northeast Brazil, respectively (distributions described in Plants of the World Online). This contrasts with the largest range sizes, such as in *Opuntia engelmannii* and *Epiphyllum phyllanthus*, which spread across multiple countries, and the smallest range sizes of extremely endemic and rare species, which can be miniscule. Cacti have smaller range sizes than most angiosperms[7,11], which are likely to further shrink due to climate change, leading to increased extinction threat[7]. Smaller ranges confer less resilience to environmental changes and habitat loss[69,70], and larger ranges are thought to elevate speciation rate via population fragmentation as the opportunity to encounter barriers and new habitats is magnified[71]. This pattern is supported by mammals and birds[72,73]. It is hard to reconcile why a pattern of faster speciation in species with larger distributions was not recovered in cacti. However, an association between faster speciation rates and smaller range sizes shapes the Brazilian Atlantic forest flora, attributed to budding speciation[74]. It is plausible that analysing geographic range size at species level in a rapidly diversifying and young lineage cannot reveal causality. Young species in radiating lineages will necessarily occupy smaller ranges than their ancestors, thus it is hard to estimate the range size at which speciation is most rapid. Nevertheless, understanding the relationship between range size and speciation should be a focus of cactus research, given the threat of climate change on range sizes[7]. We thoroughly recommend future research is undertaken with more sensitive methods than QuaSSE, especially hidden-rates models[22].

With the most comprehensive dataset for eco-evolutionary analysis yet assembled for Cactaceae, our investigations represent the state of the art for identifying the multiple drivers of cactus diversification. Although we reconstructed the most taxonomically comprehensive phylogeny yet, a fraction of nodes are relatively-weakly supported, a pattern often observed in cacti due to their recency and limited genetic diversity[13,25–27]. Future research should aim to maximise both taxonomic and molecular sampling, possibly by using the Angiosperms353[75] or Cactaceae591 probes[76].

The general trends identified here are unlikely to change significantly with increased resolution or with alternative diversification estimation methods. Incomplete taxonomic sampling and phylogenetic uncertainty could have biased diversification rate estimates, as well as the methodological choice of using BAMM to estimate speciation rates. Critics have highlighted potential sensitivities to prior distributions, which can influence the detection of speciation and extinction rate shifts[77]. Alternative tip-rate estimators include the highly accurate ClaDS[78] and the DR statistic[79]. However, analysis with ClaDS is too computationally expensive for larger lineages, and the DR statistic cannot account for incomplete and uneven taxonomic sampling. We accounted for incomplete sampling in every analysis, and our topology was congruent with previous estimates[1,2,13]. Furthermore, we implemented a highly conservative rate shift prior of one in BAMM, and limited BAMM analyses to tip-rate variation. Tip-variation is highly

robust and accurate[80] and is less sensitive to the influence of deep-time rate heterogeneity, a source of concern for BAMM[28] and diversification methods generally[81].

Recent research has begun to use machine learning to identify forces shaping macroevolution[24] and disentangle complex macroecological datasets[19]. We used it here to quantify the complexity underlying macroevolution and identify drivers of cactus diversification, finding that accounting for interactions substantially improves explanatory power, and influences the rank order of drivers. But it has also revealed how much we do not know about cactus diversification. The top three drivers identified are characters relating to climate, soil characteristics, and plant physiology, each of which have low individual predictive power. Furthermore, the complex XGBoost model reveals that ~79% of variation is yet to be explained. Interestingly, the outputs of the XGBoost models, which do not account for shared ancestry, show different relationships to QuaSSE. This indicates that our ability to test for complexity underlying the ecological factors shaping diversification need to be phylogenetically controlled, which is difficult with current methods on the scale required. Future research should aim to close these gaps, through more substantial data collection and development of statistical methods.

The cactus radiation remains one of the most iconic lineages of plants[3,8,82], especially in arid and semi-arid regions[1,2]. Cacti are the subject of intense macroevolutionary research[1,2,9,10,83–87], which has suggested a multitude of factors shaping diversification rate, some of which are correlated[2]. This has made it challenging to identify the important drivers of radiations, a problem found across the Tree of Life[22,24,68,88,89]. To address this challenge, we applied a machine learning method to account and test for complexity[23], and efficiently rank the importance of variables in an extensive dataset, revealing five key drivers. This method offers a promising direction for macroevolutionary research, by holistically and efficiently analysing dozens of interacting variables simultaneously, thus directly addressing the complexity underlying diversification. However, important drivers still require confirmation with phylogeny-controlled methods, and this framework requires significantly more intense data acquisition compared to traditional single-trait analyses. Our results suggest that cactus speciation is shaped by climate, edaphic characteristics and morphology. Further minor contributions are made by isothermality and geographic range size.

## Methods
### Supermatrix assembly and phylogenetic reconstruction
We reconstructed a phylogenetic hypothesis of Cactaceae from a supermatrix built with published genetic sequences. Orthologous loci were identified and clustered with the OneTwoTree pipeline[90]. The resulting taxonomy was checked against CITES[91] and more recent literature. We merged clusters of partial sequences with their full sequences, keeping the longest sequence when a species was present in both partial and full-sequence clusters. We added outgroup sequences from Anacampserotaceae, Portulacaceae and Talinaceae with Mafft −add[92] and visually inspected alignments for quality using SeaView[93]. Finally, we trimmed poorly aligned positions with trimAl using the command "gappyout"[94] before concatenation into a supermatrix with AMAS[95].

We reconstructed a Maximum Likelihood (ML) phylogeny using RAxML v8[96], applying a GTR model of nucleotide substitution to each locus partition, and assessing topological support by allowing bootstrapping to end automatically. In this analysis, we constrained the monophyly of several lineages to improve the likelihood calculation, after initial ML searches. These were the subfamilies (Cactoideae, Maihuenioideae, Opuntioideae), the individual genera once placed in Pereskioideae (*Leuenbergeria* and *Pereskia*, now considered paraphyletic), and the tribe Echinocereeae. We time-calibrated the final phylogeny under Penalized Likelihood (PL) using treePL[97], with stem

and crown ages given upper and lower bounds from the highest posterior probabilities estimated by[98] in a relaxed clock phylogenetic reconstruction of angiosperms. To identify the most common cross-validation optimal parameters we performed 100 priming steps in treePL. Finally, we performed cross-validation to find the smallest score, and estimated divergence times using the corresponding smooth value. More sophisticated methods of time calibration such as BEAST[99] were not possible given the size and complexity of this dataset, and we used secondary calibrations due to limited fossil availability[1].

## Data assembly

We compiled a large dataset of abiotic and biotic variables potentially contributing to variation in cactus diversification rate. Many have been previously hypothesised to drive cactus diversification (e.g., growth form, pollination syndrome and aridity), while some play a role in succulent or angiosperm diversification generally (e.g., elevation, bioclimate, plant size, topographic complexity and geographic range size). For spatial variables, we downloaded georeferenced distribution data from the Global Biodiversity Information Facility (GBIF, https://www.gbif.org/), retaining only those present in our phylogeny, and cleaned coordinates manually. We eliminated records with identical latitude and longitude, and those from cultivated specimens, botanical gardens, and purchased from greenhouses and markets. The records were visualised in QGIS to eliminate records falling outside of the coastal boundaries[100]. Finally, the distribution was corroborated manually according to the known distribution of cacti, according to botanical databases (Tropicos, APG and Plants of the World Online). All data cleaning steps were performed in Python 3.1 with the modules Pandas, Numpy and Seaborn[101–103]. GBIF data often exhibits unequal sampling across species and regions, with certain areas being underrepresented[104], potentially skewing the climatic values derived from these records. Additionally, the presence of multiple data points within a small area can lead to spatial autocorrelation, where observations are not truly independent, thus introducing bias. To minimize these spatial biases, we implemented a sampling strategy that selected only one occurrence per species within each 1 km² grid cell.

With these curated coordinates, we extracted the 19 bioclimatic variables, as well as aridity index, potential evapotranspiration and net primary productivity from CHELSA[105]. From other sources we extracted elevation and six measures of topographic complexity (slope, roughness, standard deviation of slope, standard deviation of elevation, profile curvature and tang curvature)[106], three relevant measures of soil (sand content, water and texture)[107], and biome[108]. We used median values for each species in all further analyses, except biome which is categorical, for which we used the most common biome for each species. We calculated geographic range sizes as Area of Occupancy (AOO) using the R package conR[109], using the data prior to sampling one per 1 km² grid cell, and using a cell size of 0.1×0.1 km, because of the small range sizes found in many cacti[7,11]. We are aware that this is smaller than the 2x2km cells recommended by IUCN, however, many cacti have very small geographic ranges[7,11]. By using a smaller cell size, we can better capture the variation at the smaller end, but this does reduce the estimated absolute range sizes in the small number of widespread species. To support this choice, we also calculated range size with a cell size of 2x2km and confirmed they are nearly perfectly correlated ($r = 0.997$, $p < 2.2e-16$) (Supplementary Fig. 7). We downloaded chromosome counts from the Chromosome Counts Database, using the median for each species in our analyses[110]. We built upon size (height or length), growth form and pollination data collected by refs. 2,13. Most of these data were from[12], but a small amount were from publications, or descriptions of specimens from online databases. Plant size data was mostly only available as minimum and maximum, and we recorded maximum to avoid issues with observations of incomplete, juvenile or diseased specimens. We

scored species described as "barely above ground level" or similar as zero. Following[2], we binarised growth form as globose solitary, globose caespitose or barrel form, versus arborescent, shrubby or columnar, recognising the complexity of assigning growth forms to cacti[13]. Similarly, following[2] we binarised pollination syndrome as ancestral (mellitophily, or bee-pollination) versus derived syndromes (ornithophily (birds), chiropterophily (bats) and sphingophily (moths)). We also recorded whether a species is epiphytic from[12] or species descriptions in online floras. For a full list of the 39 variables and their hypothesised role in shaping diversification rates, see Supplementary Table 2. We confirmed there are few correlations among continuous variables by plotting with corrplot[111] (Supplementary Fig. 4).

## Ranking major drivers of diversification

We estimated diversification dynamics after pruning outgroups using Bayesian Analysis of Macroevolutionary Mixtures (BAMM)[28], sampling four Metropolis coupled Markov-chains (MCMC) of 50 million generations every 5000 and discarding the first 10% as burn-in. We set priors with the R package BAMMtools[112] and implemented a conservative prior of a single rate shift. To account for imbalanced sampling, we provided sampling fractions for every genus according to CITES[91]. We assessed convergence with the R package coda[113], ensuring effective sample sizes were >200. Mean tip speciation rates were extracted with BAMMtools[112]. With mean tip speciation rates estimated by BAMM as the response variable, we assessed the relative importance of variables using the tree-based machine learning classification method XGBoost[23]. XGBoost assesses the importance of hypotheses explaining diversification by applying an adaptive learning algorithm to a set of models that are progressively better fit to the data by reweighting extreme residuals of the previous model. XGBoost provides benefits over traditional methods for this type of problem, such as phylogenetic generalised least squares which has been used to analyse family-level dynamics in plants[68]. Importantly, these benefits include relaxing the assumption of linear relationships between variables and the response, capturing complex interactions in high-dimensional datasets, and better handling of missing data and outliers. By adapting R code from[24], our XGBoost models were tuned in two steps to identify the parameters that minimise the root mean square error (rmse), for the predictive stage. First, an initial tuning step with predefined parameter combinations was performed. Following this, we refit 1000 models by randomly sampling parameters from uniform distributions bounded by the identified optimal values from stage one +/− 10%. For the final predictive model, the combination of parameters that minimised rmse was used, which resulted in a decision tree base-learner model with depth varying from 2-7. A cross-validation procedure assessed the accuracy of the XGBoost in predicting tip-rates, by randomly subsetting 80% and 20% of the data into training and testing parts, respectively. We refit the final model using the training dataset and then used the coefficients of prediction to predict tip-rates in the testing dataset. We identified significant variables with upper and lower quartiles of relative importance that did not overlap with the model threshold (1/total number of variables) and confirmed there were no correlations among these by plotting with corrplot[111] (Supplementary Fig. 5). We plotted predicted speciation rates at a range of 250 values for the significant variables, keeping other variables as their median (if continuous) or mode (if discrete). To explicitly test whether complexity better explains cactus diversification than simple models, we specified a "stump" model, the decision tree base-learner model with interaction depth =1 which does not parameterise interactions (interaction depth =1) and compared mean $R^2$.

## Sensitivity analysis

After visually inspecting BAMM-estimates tip-rates and the explanatory variables, we performed a sensitivity test. Although XGBoost is

robust to outliers, we wanted to ensure our results were robust and that we uncovered macroevolutionary trends rather than spurious relationships driven by rapidly diversifying lineages. Four genera had fast tip-rates above 0.65 (*Copiapoa*, *Gymnocalycium*, *Harrisia* and *Pilosocereus*), which could be unlinked to any overarching macroevolutionary trend across the family. These rates could lead to recovering spurious relationships with drivers. We re-ran XGBoost allowing tree-depth excluding all taxa for which tip-speciation rate exceeded 0.65, leaving 996 taxa for analysis, to verify that these rapidly speciating lineages had minimal impacts on results.

## State-dependent diversification models

We used QuaSSE[29] to analyse the impact of continuous variables identified as significant by XGBoost, after transformation to improve normality (Table 1, Figs. 2 and 3). For each variable, we fit seven models of trait-dependent diversification, in which the relationship between diversification and the variable is constant, linear, sigmoidal and hump (all with-and-without drift, except constant). Models were estimated under Maximum Likelihood in the R package diversitree[114], and we accounted for incomplete sampling by specifying the sampled fraction of described species richness. We identified the best-fitting model based on AIC, which we plotted. We produced a table comparing the best-fitting model against the null model of constant rates.

## Data availability

All Supplementary Data underlying our results are available at https://github.com/jamie-thompson/cactaceae, which includes GenBank accession numbers (in the file named "AccessionsMatrix.csv") and the entire dataset of 39 variables. Data used to make Fig. 1 is in folders "Alignment, accessions and tree" and "XGBoost and GBIF Data Assembly". Data used to make Fig. 2 is available in the folder "XGBoost relative importance tables". Data used to make Fig. 3 is available in the folder "QuaSSE model fits".

## Code availability

Sources of code used in analyses with XGBoost and QuaSSE are indicated in the methods section, and original code used for data assembly can be found at https://github.com/jamie-thompson/cactaceae. Code used to make Fig. 1 is available in ref. 64. Code used to make Fig. 2 is available in ref. 24. Code used to make Fig. 3 is in the folder "QuaSSE model fits".

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

## Acknowledgements

J.B.T. would like to thank Roger and Sue Whorrod OBE for funding a PhD studentship.

## Author contributions

J.B.T. conceptualised the study, with input from all other authors. N.K.P. supervised the work. J.B.T. collected and curated the data with assistance from G.M.K., M.V.C., and T.H.H. J.B.T. conducted analyses and visualised data. All authors discussed findings. J.B.T. wrote the initial draft, and all other authors contributed to revisions and subsequent drafts. All authors approved the final version.

## Competing interests

The authors declare no competing interests.
