## [Peer Review File · Nature Communications]

Identifying the multiple drivers of cactus diversificationReviewers' Comments:

Reviewer #1:

Remarks to the Author:

This work holds great significance not only for the field of succulent plants but also for other angiosperm groups, as it contributes to our understanding of the ecological and evolutionary processes involved in species diversification.

The manuscript makes a significant contribution to the succulent plant community by elucidating the potential factors driving the diversification of cacti groups, encompassing both biotic and abiotic influences. The discussion is highly valuable, emphasizing the multifaceted nature of these drivers and shedding light on the complex dynamics that shape plant communities. The manuscript is exceptionally well-written, presenting insightful analyses and results. Notably, the integration of sophisticated machine learning models and diversification analyses offers a promising avenue to explore previously unexplored variables that can significantly impact the diversification of these groups.

The methods section of the manuscript generally provides sufficient detail for the work to be reproduced. However, in some instances, additional details are required to ensure full reproducibility. The data analysis, interpretation, and conclusions in the manuscript demonstrate no apparent flaws. The methodology employed is up-to-date, sophisticated, and aligns with expected standards in the field, reflecting a sound approach.

The manuscript provides substantial support for its conclusions. However, it is crucial for the authors to acknowledge certain limitations associated with their raw data. Specifically, the presence of a high degree of missing data in the DNA sequence matrix, along with limited taxonomic sampling, resulted in a phylogenetic tree with low support and critical taxon-sampling gaps. It is important to consider these limitations and their potential impact on the overall reliability and validity of the results, as all diversification analyses are based on this tree. While these limitations do not necessarily prohibit publication, addressing them explicitly and discussing their potential implications in the manuscript would enhance transparency and strengthen the study.

Major comments:

1- The macroevolutionary analysis requires a well-supported phylogeny to estimate speciation and diversification rates. However, most of the nodes of the inferred phylogenetic tree have low (<70%) bootstrap support. How could these low-supported nodes impact the results of diversification analysis? By acknowledging the presence of low-supported nodes and discussing their potential impact, the authors can provide a more comprehensive assessment of the uncertainties and limitations associated with the diversification analysis, thereby enhancing the transparency and reliability of the study's results.

2 - The authors incorporated sampling fractions information in their diversification analyses to address imbalanced taxon sampling. However, it is important to note that they did not provide specific details regarding the main taxonomic and geographic sampling gaps, including the sampleProbs file used in the Bamm analysis. This lack of information hinders the evaluation of potential impacts arising from these sampling gaps on the main results. For instance, it raises the question of whether the underrepresentation of South American taxa in the phylogeny, as opposed to cacti from the Northern Hemisphere, could affect the recovery of the atypical latitudinal gradient pattern in diversification rates. Including and discussing these details would contribute to a more comprehensive understanding of the potential limitations and caveats associated with the study's findings.

3 - The authors computed the average abiotic variable values for each occurrence point obtained from GBIF data and employed them as feature values for each species in their machine learning and diversification models. However, this methodology could significantly influence the outcomes in cases of species records exhibiting a bias towards extensively explored geographic regions. To mitigate this issue, I recommend that the authors reduce the occurrence points to one record per grid for

calculating median values. Furthermore, it is advisable to incorporate various summarization parameters beyond the median, such as mean, mode, maximum and minimum boundaries, as well as standard deviation, when utilizing statistics summarization in machine learning models.

4 - (LI. 495-498). Please, provide a more detailed description of the calibrated nodes.

Minor comments:

6 - LI. 59-60. Please, include birds in the cactus pollinators list.

7 - LI. 529-536. I suggest the authors provide an updated database of biotic data, including new values not presented in Hernández-Hernández et al. 2011.

8 - LI. 128-130. How might the missing data for biotic variables impact the machine-learning models?

9 - LI 156-158 and 221-222: How could the mentioned limitations of QuaSSE models affect the results leading to the conclusion that diversification peaks around the Tropic lines?

10 - L. 224. Please, correct it to "North America Southwest".

11 - L. 241. There is a more updated reference about diversification rate shifts in the genus *Pilosocereus* that could be cited here (Romeiro-Brito et al. 2023, <https://doi.org/10.1002/ajb2.16134>).

12 - LI. 293-294. I would expect slow-growing organisms to reach reproductive age later than fast-growing organisms, resulting in the former having a longer generation time. Please, clarify this sentence or correct it to "slow growing increasing generation time."

13 - LI. 407-409. - Lines 407-409: Why would highly mobile pollinators be more efficient at promoting reproductive isolation? This sentence requires clarification.

14 - LI. 428-430. The findings of Leão et al. (2020) regarding the impact of budding speciation on range size are specific to the flora of the Atlantic Forest, a wet biome in Brazil, rather than encompassing the entire 'Brazilian flora'. Please consider revising this sentence accordingly.

Reviewer #2:

Remarks to the Author:

Thompson et al. - Identifying multiple drivers of cactus diversification. In this manuscript the authors are attempting to uncover the underlying processes that have shaped cactus diversification, a very noteworthy endeavor and one that is certainly complex. I have made editorial comments below and also have some major questions that I think should be addressed that might help better convey what was done in the manuscript (see below).

Intro line 44, what is meant by modified spines? Perhaps the authors are referring to modified leaves in the form of spines?

In the intro, it is not clear why aridification would not have had an influence on diversification. In the species-poor examples given, those are from areas which are in general less arid, and the other larger lineages given as examples are from more arid zones in general (Cactoideae and the Mammilloid clade)... so the authors' logic seems inherently flawed.

First paragraph: The authors mention Pereskioideae. I assume this is meant to take into account both clades, which are now commonly referred to as *Leuenbergeria* and *Pereskia* and have been shown since 2005 to form a paraphyletic group, thus "Pereskioideae" is not a clade and should not be referred to as a natural group-Guerrero et al. 2018 explain this-please revise. The tree generated here also reflects this paraphyletic group. So, if Pereskioideae is to be used, it should be just for the *Pereskia* clade leaving out *Leuenbergeria*. That should be made clear, and would include less than 19 species.

Line 60, The authors state that diversification rates are fastest in species with larger growth forms, but that is apparently not the case. The mammilloid clade has been shown to have the fastest growth rates in Cactaceae (see Breslin et al. papers cited here).

Line 74, the authors state that growth form is dependent on pollination syndrome, but there seems to be no real pattern associated with growth form and pollination syndrome in cacti. Hummingbird pollination, for example, is found in trees, shrubs, barrel-shaped species, as well as in the Mammilloid clade and epiphytes. The same could be said for insect or bat pollination, so this statement needs some clarification.

Line 56, cacti should not be capitalized

Line 60, birds also are a major pollinator in cacti and could be added to this list.

Line 63, cactus should not be capitalized, ditto in lines 68, 73 and throughout

Line 155, should be cacti not cactus

Line 187, I think this will need more explanation. It is said that species with a derived chromosome count (thus polyploids) speciate faster than those with diploid numbers. I think this is likely a misinterpretation of the data. Certainly polyploidy likely leads to high rates of speciation, but there are very few instances across Cactaceae where speciation appears to have occurred from polyploid ancestors, which is what this sentence seems to be implying. In general, polyploids appear to be mostly derived from diploid ancestors across Cactaceae (the authors cite Castro et al. and could dig into this more from the results in that paper). There perhaps is one small clade of nine species that actually has speciated after polyploidization... Otherwise, as far as is known, this is super rare across the family.

Line 307, the authors mention species of Mammillaria with modified leaves. All cacti have modified leaves (spines), so it is unclear what is meant here. Also, Mammillaria have no noticeable photosynthetic leaves on any part of the plant, because they have been so heavily modified into spines. So, this should be clarified.

Line 336, the authors mention that soils with high sand content are likely to be deserts. This is not clear at all. Sandy soils are not the foundation for deserts, as this sentence reads. Rather sandy soils may lead to edaphically dry areas, regardless of whether or not they are located in deserts. Sandy soils are global and do not necessarily correspond to deserts. Check the Guianan Shield, Amazonian white sand forests, and the Southeastern United States as examples in the Americas...

The literature on Cactaceae is super extensive, so the authors should try to avoid overciting their own work. For instance, in the conclusions the authors state, "Cacti are the subject of intense macroevolutionary research, revealing a multitude of factors shaping diversification rate, some of which are correlated (Hernández-Hernández et al., 2014)." One paper is not intense macroevolutionary study...

Suppl. Figure 1, It would be good to provide a table to show where the rate shifts have occurred or label this with the clade name on the figure. Otherwise, the figure is hard to evaluate, other than seeing rate shifts have occurred. This is important given the authors are suggesting certain features, such as growth form, are associated with rate shifts.

Suppl. Figure 2, What is C? It is not listed in the legend.

Figure 1, the cholla and the Opuntia photos are located in the wrong clades based on their phylogeny. Please check the Eriosyce clade to make sure the photo used is an Eriosyce. It is not obvious from the figure.

Figure 5, legend, arboreal generally refers to something living in a tree (often animals). I think the more appropriate term here would be arborescent, when referring to growth form.

Given the controversy around the use of BAMM, it could be worthwhile for the authors to discuss their use of the tool.

The Chromosome Counts Database has not been updated with any new records for Cactaceae, since the mid-2000s, so it might not be the most reliable source for counts. The authors also used the median count for each species in their analysis. Many studies have shown that multiple species are often represented by numerous ploidal levels often found within "one" species, so it may perhaps be more accurate to treat multiple counts separately (although not as easy).

The authors used Hunt (2016) for their taxonomy, which is now outdated. Why not use a more recent and expertly curated (from the Cactaceae community) list for the family, such as Korotkova et al. 2021?

It would be good for the authors to describe how they measured growth size in Cactaceae, especially given that they found it to be significant. What is meant by small vs. large and intermediate, as given in the results and discussion. I presume that height and length were synonymous given the description in the methods, but this is not totally clear. Are there actual values associated with these categories, or are the criteria subjective? One person's large might be another person's intermediate... This could be greatly clarified here, instead of citing other papers. This data also does not appear to have been made available on Github, so it is hard to evaluate.

Pilosocereus is given as an example from the Caribbean of a rapidly evolving clade. In general the Caribbean is not super species rich, as compared to continental high diversity areas, as the authors clearly state. That also goes for Pilosocereus, which is much more species rich in South America, for example. The tree used here only has three species of Pilosocereus from the Caribbean, so it is a little baffling how that clade is highlighted for rapid speciation in the Caribbean region. Could this be a misinterpretation of the results? Looking at the map of points, the Caribbean is quite red, but only the Pilosocereus clade is really represented in the phylogeny (Leptocereus only has a couple of taxa, Melocactus contains only one Antillean species in the tree). So these larger radiations in the Antilles are essentially not represented. Thus, it seems that the red points representing diversification in the Caribbean must be lots of individual points for just three species of Pilosocereus? This is confusing, and I don't think it reflects reality. Perhaps I don't fully understand the methodology of how the points were included here. It would be nice to see some clarification.

In general, the manuscript appears to be lacking in clear methodology, some misinterpretation or misrepresentation of the literature and basic morphological features of cacti (and not broad enough coverage of the literature) and results, and does not present a terribly convincing argument that the authors have found novel patterns here. In the conclusions, I am left wondering what was discovered in this manuscript, other than a latitudinal gradient corresponding to diversification rates. Across the Americas cacti generally started to diverge significantly in the Miocene and then most of the overall diversity seems to have been generated in the Pliocene and into the Pleistocene epochs, so why would only lower latitudes produce this pattern in this dataset? This is not well explored. When looking at the scatterplot for plant size in Figure 1, the vast majority of points are all in the lower size range, then with a few outliers at the higher end, so are larger taxa really diversifying faster? If I look at the red parts of the tree (faster diversification), those all correspond to small plants in the phylogeny (Mammillaria, Echinocereus, Rhipsalis, Eriocyce, etc.) and which are not all at lower latitudes (i.e., Mammillaria and Echinocereus are most abundant in Mexico). So, if there truly a latitudinal pattern here? See also my comment above about potentially biased data with Pilosocereus over-representation in the Caribbean.

Reviewer #3:

Remarks to the Author:

General comments

The manuscript by Thompson et al. investigates the relative importance of a range of abiotic and biotic variables in influencing the rates of speciation on the iconic botanical family Cactaceae. By applying phylogenetic comparative methods, in combination with boosted regression tree analyses, the authors describe a complex set of factors as potential drivers of speciation, particularly latitude and plant size. Overall, the manuscript is well written, and I recognize the effort that went into creating a comprehensive dataset of variables. However, I did find some relevant methodological shortcomings that, in my opinion, preclude acceptance of the manuscript as it currently stands. As you will see from my comments below, my main concern relates to the interpretation of the effect of some variables, in addition to an overall shallowness in method description and data exploration. Please, do not interpret these comments as a discouragement, but as recommendations for the improvement of the work.

Major points

My first major point relates to the use and finding of latitude as a predictor for speciation rates. Latitude is of course correlated with species richness in most taxa in such way that one of the most examined macroecological patterns is the latitudinal diversity gradient. However, as a predictor, what does latitude refer to? This point is relevant because as stated multiple times in the manuscript, latitude is one of the main "drivers" of Cactus diversification. But how can latitude "drive" diversification rates? Latitude itself does nothing that makes any biological sense, but variables correlated with latitude might do (e.g. temperature, humidity, energy, seasonality). Beyond the potential effect related to the time-for-speciation hypothesis (i.e. the tropics have been geologically more stable through time, allowing more time for species to accumulate), I see no other way that latitude could be a "driver" of evolutionary rates. In fact, the results clearly point to that, when the authors claim that: "...we find latitude is the strongest predictor in both XGBoost models, suggesting a primary role of latitude over covariates...after accounting for interactions in the complex model the relative importance of several climatic covariates is reduced" (lines 259-253). This is perhaps an indication that latitude is explaining more variability because it correlates with several other climatic covariates, and therefore, has a much better power for separating residuals in the boosted regression tree analysis.

Having said that, I wonder what happens to your results if you remove latitude and only keep the climatic variables in the model? I would also suggest looking at a covariance structure between explanatory variables, to check if there is a high degree of collinearity between them (maybe through the variance inflation factor - VIF). This would potentially help to disentangle what variables correlated with latitude are more likely influencing speciation. This point is also relevant because of the large number of variables included in the model. The authors claim that 11 variables were identified to "significantly contribute to the diversification of Cactaceae" (lines 22-24). However, upon closer inspection (Fig. 2), most of these variables have a relative importance of less than 5% in the final model. This only represents 1.1% of total variance explained, considering that their best model only explains a total of 22% of the variability in rates. This is quite a low percentage for many of the factors that have been discussed as potential "drivers" of diversification. I agree with the authors that modelling evolutionary rates is complex, and their study is a testament of that. However, I believe that focusing the discussion on factors that were identified as "above chance expectation" might not be the best strategy in this case. This makes the discussion unnecessary long and complex, leaving the impression that almost anything could be contributing to higher diversification in the group, which might not necessarily be the case given the low explanatory power. Hence, my suggestion here is to try and narrow down the number of variables analyzed (maybe with the suggestion above) before including them in the final boosted regression tree model. This will hopefully provide more resolution and a better final model.

Now, back to the latitude topic, by looking at Fig. 4, it seems like the trend of higher tip-speciation is found in the Caribbean for +25° and in regions of potential endemism in South America (Brazilian Cerrado?) for -25°. Have the authors thought about including endemism and species richness, for example, as potential explanatory variables for speciation rates? This would potentially give more context to "latitude" with more relevant ecological predictors. In fact, I cannot find anywhere in the

manuscript a list of all the 39 variables, so maybe the authors should consider including a supplementary table with the variables and their hypothesized effect. This could also help to narrow down the number of variables included in the model, after critically evaluating the role of each one. Regarding the relationship with size, although the authors find that "An inverted modal relationship between Cactus size and speciation rate is best supported, with fastest rates in the smallest and largest Cacti, and reduced rates in intermediate sizes (Figure 3)", this does not seem to agree with the results for tip-speciation (Fig. 1). The relationship of tip-speciation with size seems to suggest that only smaller species tend to have really high speciation, which is interesting but completely ignored in the manuscript. This is particularly important because the authors discuss "larger forms speciating rapidly, contrary to much research on size evolution across the Tree of Life." (lines 473-474). In my opinion, this point should be further explored because the tip-rates suggest that your patterns are not contradicting expectations, as suggested.

Finally, I was also worried about the overall lack of support for the phylogenetic tree used in the manuscript. The authors suggest that the phylogeny was "moderately well-supported", but the reality is that the majority of nodes (almost 70%) had very low support values. In addition, the time-calibration step only used two points of calibration, which sounds like very little information for such large phylogeny. All of this is incredibly relevant, given that all downstream analyses are dependent on this phylogeny, which has the potential for error propagation if the support is so low and the calibration is uncertain.

Other Points

Overall, I felt like the methods section could be improved. For example, there is only one sentence explaining how you accounted for interactions in the boosted regression tree analysis, despite this being an important aspect of the manuscript. I suggest being more specific in the methods to allow for clarity and reproducibility.

Line 151: I failed to find your sensitivity analyses excluding outliers in the supplementary data. Please double check.

Line 1149: "best-fitting" instead of "best-fitted".

Figure 5: Given that HiSSE had a better support, shouldn't this plot show the HiSSE results instead? This also applies for the results section. If HiSSE had a better support, it means that unobserved traits are more influential to diversification rates than these two, which is also indicated by the XGBoost results.

I hope this review was helpful.

Dear reviewers,

Thank you for the helpful comments we received on our manuscript "Identifying the multiple drivers of Cactus diversification" (NCOMMS-23-26213A), which was invited for a major revision. Please see our substantial revision of the manuscript (NCOMMS-23-26213A R1). We have addressed each of the major corrections, including a complete re-sampling of the spatial data and rerunning all analyses based on this better-curated data. We have also improved our provision of data, and completely renewed the Supplementary Materials. We hope you will agree that addressing the issues raised by reviewers has produced a substantially stronger manuscript. Below, please see how we addressed these comments.

Reviewer #1 (Remarks to the Author):

This work holds great significance not only for the field of succulent plants but also for other angiosperm groups, as it contributes to our understanding of the ecological and evolutionary processes involved in species diversification.

The manuscript makes a significant contribution to the succulent plant community by elucidating the potential factors driving the diversification of cacti groups, encompassing both biotic and abiotic influences. The discussion is highly valuable, emphasizing the multifaceted nature of these drivers and shedding light on the complex dynamics that shape plant communities. The manuscript is exceptionally well-written, presenting insightful analyses and results. Notably, the integration of sophisticated machine learning models and diversification analyses offers a promising avenue to explore previously unexplored variables that can significantly impact the diversification of these groups.

– Thank you for these comments, we greatly appreciate them.

The methods section of the manuscript generally provides sufficient detail for the work to be reproduced. However, in some instances, additional details are required to ensure full reproducibility. The data analysis, interpretation, and conclusions in the manuscript demonstrate no apparent flaws. The methodology employed is up-to-date, sophisticated, and aligns with expected standards in the field, reflecting a sound approach.

– Thank you for these comments on our analytical approaches. We have edited some areas of the methods section, regarding phylogenetic inference (lines 426-431), spatial-data curation (lines 450-458), and XGBoost hypothesis testing (526-533).

The manuscript provides substantial support for its conclusions. However, it is crucial for the authors to acknowledge certain limitations associated with their raw data. Specifically, the presence of a high degree of missing data in the DNA sequence matrix, along with limited taxonomic sampling, resulted in a phylogenetic tree with low support and critical taxon-sampling gaps. It is important to consider these limitations and their potential impact on the overall reliability and validity of the results, as all diversification analyses are based on this tree. While these limitations do not necessarily prohibit publication, addressing them explicitly and discussing their potential implications in the manuscript would enhance transparency and strengthen the study.

– Thank you - we agree with these sentiments, and have discussed them thoroughly in lines 360-387, in a new “Future directions” discussion paragraph. We have discussed weak node support, and how this could be improved. In addition, we have discussed incomplete sampling and the ways we accounted for this. We have also discussed the issues associated with diversification studies, revealed by Louca and Pennell (2020), and how our approaches mitigate these concerns.

Major comments:

1- The macroevolutionary analysis requires a well-supported phylogeny to estimate speciation and diversification rates. However, most of the nodes of the inferred phylogenetic tree have low (<70%) bootstrap support. How could these low-supported nodes impact the results of diversification analysis? By acknowledging the presence of low-supported nodes and discussing their potential impact, the authors can provide a more comprehensive assessment of the uncertainties and limitations associated with the diversification analysis, thereby enhancing the transparency and reliability of the study's results.

– Thank you, we did not go into depth on this matter because lead author Thompson believed it was common knowledge that cactus phylogenies typically have low support, on account of their low sequence divergence and young clade age. But we recognise this was naive to assume, and have now added discussion points, with an explicit caveat about their impacts on our diversification estimates and results. We have also explicitly referenced other studies which have also recovered low phylogenetic support (lines 365-367).

– While we strongly agree that low support can give spurious diversification-rate estimates, we still believe we have provided a strong framework for cactus diversification analyses.

– Macroevolution studies typically have to prioritise high-resolution or high-taxonomic coverage, by sampling the genome widely for a few representative species, or sampling relatively few loci for as many species as possible. In studies of such rich lineages as

Cactaceae, the latter has to be prioritised because it is currently impossible to do phylogenomic analyses for >1,500 species.

– I have discussed this issue at length with Prof Dan Rabosky (the author of BAMM) and he describes how “the existence of a cluster of branches with poor topological resolution is often a signal of rapid diversification at that point” (from an email correspondence). Cacti have lots of pockets of low topological uncertainty distributed throughout their phylogeny, especially in younger and rapidly-diversifying lineages. Many lower-level taxonomic groups are poorly resolved, yet, due to their rapid branching, we can be confident of the existence of a rate shift.

– Furthermore, this is why we chose to use BAMM to estimate the species-specific speciation rates, instead of the competing DR statistic. BAMM discretises rates into “regimes” in which the descendants of a node with a rate shift are characterised by similar diversification dynamics. The DR statistic is more accurate when a topology and terminal branch lengths are “perfect”, because it does not discretise and estimates species-specific rates. However, it increases variance of estimates based on terminal branch lengths, which is not ideal in the case of a tree shaped by several poorly-resolved recent radiations. As we do not perfectly know the true branching order in younger cactus lineages, species-specific speciation rates beyond the discretisation provided by BAMM would be inaccurate.

– We know there are many recent radiations in cacti leading to poor topological support, and can identify rate shifts leading to these branches, but not to a precision beyond that. BAMM “smoothes” tip-rates of descendants, reflecting this uncertainty.

– Another quote from Rabosky for illustration (from an email correspondence). “There are topological and branch length combinations where we can have extremely high confidence that a shift occurred if the topology itself is highly uncertain. For example: consider a clade that experiences a massive spike in speciation followed by a slowdown. We might have a really hard time resolving the topology, and there might be no confidence at all in any particular pattern of branching within that focal subclade. Yet, we could still be confident that a shift occurred on the branch leading to the MRCA of the subclade.”

2 – The authors incorporated sampling fractions information in their diversification analyses to address imbalanced taxon sampling. However, it is important to note that they did not provide specific details regarding the main taxonomic and geographic sampling gaps, including the sampleProbs file used in the Bamm analysis. This lack of information hinders the evaluation of potential impacts arising from these sampling gaps on the main results. For instance, it raises the question of whether the underrepresentation of South American taxa in the phylogeny, as opposed to cacti from the Northern Hemisphere, could affect the recovery

of the atypical latitudinal gradient pattern in diversification rates. Including and discussing these details would contribute to a more comprehensive understanding of the potential limitations and caveats associated with the study's findings.

– In our original BAMM and current analyses, we mitigated the impacts of taxonomically-imbalanced and incomplete sampling with a genus-level sample fraction file, available in our Github repository (in the updated folder “Alignment, accessions, sampling fraction file and tree.zip”). Though this approach does not explicitly address imbalanced spatial sampling, it accounts for incomplete sampling, especially given that we sampled the vast majority of genera with at least one representative species.

– As recommended by both reviewers, we have addressed imbalanced spatial sampling in the revised manuscript by computing one occurrence per-species-per-grid-cell.

– As recommended by reviewer 3, we have also removed latitude as a predictive variable in the model and focussed on the explanatory power of covariates. Given that it is difficult to account for such broad-scale issues as imbalanced sampling between Northern and Southern America, we believe our decision to remove latitude is justified, thereby removing the risk of over-inflating the LDG pattern recovered in our GBIF data. Reassuringly, factors covarying with latitude are recovered as strong, revealing the actual mechanisms behind diversification.

– These steps have not removed spatial bias in their entirety, this is well-known to be impossible. But by reducing our sampling to one per grid cell, and removing latitude as a predictor, we believe we have mitigated the primary biases, and potential for false LDG patterns.

3 – The authors computed the average abiotic variable values for each occurrence point obtained from GBIF data and employed them as feature values for each species in their machine learning and diversification models. However, this methodology could significantly influence the outcomes in cases of species records exhibiting a bias towards extensively explored geographic regions. To mitigate this issue, I recommend that the authors reduce the occurrence points to one record per grid for calculating median values. Furthermore, it is advisable to incorporate various summarization parameters beyond the median, such as mean, mode, maximum and minimum boundaries, as well as standard deviation, when utilizing statistics summarization in machine learning models.

– Thank you, we had not considered the impact of this geographic sampling issue. We have redone our analyses by reducing the occurrence points to one record per 1km grid cell (per species), before calculating the median for further analyses.

– Furthermore, we have also replaced the 19 bioclimatic variables originally from worldclim, and the aridity index and potential evapotranspiration from CGSIAR, with CHELSA data (Karger et al., 2017 *Sci. Data* doi: 10.1038/sdata.2017.122). CHELSA is more up-to-date and is known to have much higher resolution and accuracy than wordclim and CGSIAR. We have also included net primary productivity, as recommended by a colleague, although this was not significant according to XGBoost.

– We appreciate the concern with only using median values for the XGBoost and following QuaSSE models, and the value of including other summarisation methods. However, we did not replicate XGBoost or QuaSSE analyses with mean, mode, max, min or st dev.

–This is partially due to the prohibitive computational requirement (QuaSSE models on a phylogeny this size can take over a week each to fit on our server), but this is also because using the median is robust to outliers. This is particularly important for using data derived from GBIF coordinates, which may suffer from identification issues, or species found outside their natural ranges (although we did account for this somewhat by curating the data against described distributions). It is likely that these issues are elevated in cacti, in which many species are difficult to distinguish in field settings.

– We believe it would be inappropriate to analyse minimum and maximum values with XGBoost. The biotic data were largely discrete, with the exception of plant height, for which minimum and maximum values were provided by Anderson (2001). Using minimum values for size would be inaccurate, as there is potential that these values are from incomplete, juvenile or diseased individuals (lines 480-483). While maximum values also potentially introduce bias, this is much lower than minimum values.

– Analysing minimum and maximum values for variables derived from GBIF would likely introduce significant bias. In addition, analysing mean values would also introduce bias. Standard deviation would be too sensitive to species for which there are fewer geographic records than others. We believe that analysing median values of GBIF-derived variables represents a good middle-ground that is not as sensitive to outliers and other GBIF-specific issues (e.g. misidentified species, an important problem for Cactaceae research).

4 – (Ll. 495-498). Please, provide a more detailed description of the calibrated nodes.

– We have provided further detail here, by stating “In this analysis, we constrained the monophyly of several lineages to improve the likelihood calculation, after initial ML searches. These were the subfamilies (Cactoideae, Maihuenioideae, Opuntioideae), the genera once placed in Pereskioideae (*Leuenbergeria* and *Pereskia*, now considered paraphyletic), and the tribe Echinocereaeae.” (Lines 426-431).

Minor comments:

6 - Ll. 59-60. Please, include birds in the cactus pollinators list.

– Thanks for pointing out this oversight, we have corrected it.

7 - Ll. 529–536. I suggest the authors provide an updated database of biotic data, including new values not presented in Hernández-Hernández et al. 2011.

– These data are provided in the associated GitHub repository (<https://github.com/jamie-thompson/cactaceae/>), and we intend to submit this dataset to a journal such as “Scientific Data” or “Data in Brief”.

8 - Ll. 128-130. How might the missing data for biotic variables impact the machine-learning models?

– We acknowledge that missing data, especially in large datasets, is a difficult challenge in ecological research. However, XGBoost is designed to handle missing data (<https://xgboost.readthedocs.io/en/stable/faq.html>) and performs well with under these circumstances (Aydin and Ozturk, 2021

https://www.researchgate.net/publication/350135431_Performance_Analysis_of_XGBoost_Classifier_with_Missing_Data).

– When XGBoost encounters a missing value for a variable, it learns the best direction to assign these missing values during training. This is achieved by allowing the model to 'learn' whether missing values should be grouped with a specific split of the data or another, based on what improves the model's performance. This feature of XGBoost is particularly beneficial in ecological datasets, where missing data is a common problem due to data collection constraints.

– Missing data is unlikely to have shaped our results beyond minor details. Furthermore, the most important biotic traits (at least *a priori*, based on previous hypotheses) have strong coverage (e.g. growth form, 100%).

9 - Ll 156-158 and 221-222: How could the mentioned limitations of QuaSSE models affect the results leading to the conclusion that diversification peaks around the Tropic lines?

– We have since removed latitude as recommended by reviewer 3, in order to focus on drivers of diversification rates.

10 - L. 224. Please, correct it to “North America Southwest”.

– We have corrected this.

11 - L. 241. There is a more updated reference about diversification rate shifts in the genus *Pilosocereus* that could be cited here (Romeiro-Brito et al. 2023, <https://doi.org/10.1002/ajb2.16134>).

– Thank you. But we have since removed analyses of latitude, due to the covariation with explanatory variables (e.g. climate), so have not included this.

12 - Ll. 293-294. I would expect slow-growing organisms to reach reproductive age later than fast-growing organisms, resulting in the former having a longer generation time. Please, clarify this sentence or correct it to “slow growing increasing generation time.”

– This was an oversight, thanks for noticing it. We have corrected it.

13 - Ll. 407-409. - Lines 407-409: Why would highly mobile pollinators be more efficient at promoting reproductive isolation? This sentence requires clarification.

– We have provided clarification on this, by stating that “This greater dispersal distance can promote reproductive isolation by facilitating the spread of alleles over larger geographic areas, which can lead to local adaptations in the absence of gene flow (Xu et al., 2012). Both mechanisms can accelerate the divergence of populations and lead to speciation.” (Lines 297-301).

14 - Ll. 428-430. The findings of Leão et al. (2020) regarding the impact of budding speciation on range size are specific to the flora of the Atlantic Forest, a wet biome in Brazil, rather than encompassing the entire 'Brazilian flora'. Please consider revising this sentence accordingly.

– We have corrected this.

Reviewer #2 (Remarks to the Author):

Thompson et al. - Identifying multiple drivers of cactus diversification. In this manuscript the authors are attempting to uncover the underlying processes that have shaped cactus diversification, a very noteworthy endeavor and one that is certainly complex. I have made editorial comments below and also have some major questions that I think should be addressed that might help better convey what was done in the manuscript (see below).

– Thank you, we seriously appreciate your comments which have made the manuscript much better. Especially surrounding our references to the wider literature, our silly oversights and grammatical errors, and your thorough knowledge of cactus evolution.

– Several of your points are extremely helpful (e.g. polyploidy, growth form and pollination syndrome), but have since become redundant since re-running the analyses with better geographic sampling and without latitude. We have since found, with better sampling methods, better climatic data, the removal of latitude, and a more-stringent assessment of “significance” in XGBoost, that chromosome count and growth form are no longer significant.

– All of your comments are very helpful, and we have replied to all of them, including the CCDB point.

Intro line 44, what is meant by modified spines? Perhaps the authors are referring to modified leaves in the form of spines?

– Thank you for noticing this oversight, we have corrected it to “modified leaves often taking the form of spines”.

In the intro, it is not clear why aridification would not have had an influence on diversification. In the species-poor examples given, those are from areas which are in general less arid, and the other larger lineages given as examples are from more arid zones in general (Cactoideae and the Mammilloid clade)... so the authors’ logic seems inherently flawed.

– Thank you for raising this oversight. These examples were poor choices and were originally provided to illustrate the phylogenetic imbalances. We have amended this by removing these examples entirely, and instead provided examples of macroevolutionary research, including by our lab, into timings of major radiations, and the relationship between aridity index and succulent diversification rates (lines 51-54, 56-57).

First paragraph: The authors mention Pereskioideae. I assume this is meant to take into account both clades, which are now commonly referred to as *Leuenbergeria* and *Pereskia* and have been shown since 2005 to form a paraphyletic group, thus “Pereskioideae” is not a clade and should not be referred to as a natural group-Guerrero et al. 2018 explain this-please revise. The tree generated here also reflects this paraphyletic group. So, if Pereskioideae is to be used, it should be just for the *Pereskia* clade leaving out *Leuenbergeria*. That should be made clear, and would include less than 19 species.

– Yes, using the word Pereskioideae was our perhaps-clunky way to describe genera *Leuenbergeria* and *Pereskia*, while trying to maintain simplicity to illustrate the point about

phylogenetic imbalance. We recognise that these genera are paraphyletic throughout analyses, and have reflected that by rephrasing this as “For example, subfamily Maihuenioideae, Pereskioideae, *Leuenbergeria* (once considered a member of Pereskioideae)”. (Lines 429-430).

Line 60, The authors state that diversification rates are fastest in species with larger growth forms, but that is apparently not the case. The mammilloid clade has been shown to have the fastest growth rates in Cactaceae (see Breslin et al. papers cited here).

- It may be counterintuitive that diversification rates are fastest in species with larger growth forms, although this has been demonstrated previously for subfamily Cactoideae (Hernandez-Hernandez et al., 2014, *New. Phyto.*). This was explained by these authors as the result of coevolving with derived pollination syndromes.
- Our results extend this known relationship between elevated diversification rates and larger growth forms to the subfamilies of Cactaceae which were not sampled by Hernandez-Hernandez et al. (2014).
- Breslin et al. (2022, *Am. J. Bot.*) analysed phylogenetic relationships of 89 members of the Mammilloid clade, and found that diversification rates are higher than we recovered for these species.
- While the Mammilloid clade is certainly an important and large lineage within cacti, and Breslin et al. analysed much denser genomic data than we could, given the family-wide scope of our analyses, we caution that our results are looking across the diversity of all cacti. Any pattern of elevated diversification rates of Mammilloid species, which have smaller growth forms, may be washed out by the general trend of larger growth forms diversifying rapidly (e.g. Pachycereeae, Hylocereeae and Trichocereae).
- Furthermore, discrepancies between results of our and Hernandez-Hernandez et al. (2014) analyses, and analyses by Breslin et al. are likely to be due to the uncertainty associated with divergence times within Cactaceae. Breslin et al. recovered a younger divergence time for the Mammilloid clade than we did, leading to more-rapid diversification rates.
- Any time-calibration in Cactaceae, or other plant clades without a fossil record, should be taken with a grain of salt. This is the reason why we focussed our Machine Learning analysis on tip-rates, and used algorithms for which results can be considered relative rather than absolute (e.g. QuaSSE).

– There are different merits and pitfalls to our and Breslin et al.’s analyses, and all offer valuable insights into cactus evolution.

- While Breslin et al. certainly sampled the plastid genome more densely than we did and were able to use Bayesian methods such as BEAST, due to their focussed taxonomic sampling, across-family rate heterogeneity may have been lost, which impacts divergence times especially when no useful fossils are available. Furthermore, Breslin et al. used estimated divergences from a paper published in 2011 as secondary calibrations (Arakaki et al., *PNAS*).
- We sampled the plastid genome less densely (although 15 plastid loci made it into our supermatrix, which is quite substantial compared to many analyses). But we also sampled three nuclear genes, which can help improve angiosperm phylogenetic inferences (Zhang et al., 2012 *New Phyt.* <https://doi.org/10.1111/j.1469-8137.2012.04212.x>), and divergence times (Zeng et al., *Nat Comms.* <https://doi.org/10.1038/ncomms5956>). Furthermore, we sampled taxonomic diversity more broadly (1,063 species across 125 genera), which improves our understanding of rate-heterogeneity and subsequent divergence estimates. We also used up-to-date divergence estimates from an angiosperm-wide time-tree (Ramirez-Barahona et al., 2020, *Nat. Eco. Evo.*).

– With this in mind, we believe that there is sufficient evidence to state that larger growth forms diversify more rapidly than smaller growth forms across Cactaceae generally, although there is certainly uncertainty with specific lineages such as the Mammilloid clade, as revealed by recent phylogenomic studies such as Breslin et al. (2022). It is likely that, as we discuss in our MS, complexity is involved at every level of driving macroevolutionary dynamics. Different combinations of forces are responsible for elevated diversification in different groups, although we are still able to find commonalities across the family.

Line 74, the authors state that growth form is dependent on pollination syndrome, but there seems to be no real pattern associated with growth form and pollination syndrome in cacti. Hummingbird pollination, for example, is found in trees, shrubs, barrel-shaped species, as well as in the Mammilloid clade and epiphytes. The same could be said for insect or bat pollination, so this statement needs some clarification.

– While some pollinators are generalists and visit many lineages, Hernandez-Hernandez et al. (2014) robustly demonstrate a coevolution between derived pollination syndromes and larger growth forms in subfamily Cactoideae. This pattern is also recovered in our data across the non-Cactoideae subfamilies and lineages, although we did not present this finding in our data. We are happy to provide the results if requested. We have amended the statement by adding “derived”, which clarifies that it is not one single group of pollinators that coevolves with growth form (line 81).

Line 56, cacti should not be capitalized

– Thanks, we have corrected this.

Line 60, birds also are a major pollinator in cacti and could be added to this list.

– This was an oversight, thanks for pointing this out. We have corrected this.

Line 63, cactus should not be capitalized, ditto in lines 68, 73 and throughout

– Thank you, we have gone through the manuscript and edited instances of “Cacti” and “Cactus” to “cacti” and “cactus.”

Line 155, should be cacti not cactus

– Thanks, we have corrected this.

Line 187, I think this will need more explanation. It is said that species with a derived chromosome count (thus polyploids) speciate faster than those with diploid numbers. I think this is likely a misinterpretation of the data. Certainly polyploidy likely leads to high rates of speciation, but there are very few instances across Cactaceae where speciation appears to have occurred from polyploid ancestors, which is what this sentence seems to be implying. In general, polyploids appear to be mostly derived from diploid ancestors across Cactaceae (the authors cite Castro et al. and could dig into this more from the results in that paper). There perhaps is one small clade of nine species that actually has speciated after polyploidization... Otherwise, as far as is known, this is super rare across the family.

– Thank you, this is very interesting and perhaps we did not appreciate the depth of investigation required to make these statements. However, during major revisions including by resampling the spatial data, and by removing latitude as a predictor, chromosome count has become non-significant as the predictive power of other variables have grown. All discussion around chromosome count and polyploidy has therefore been removed.

Line 307, the authors mention species of Mammillaria with modified leaves. All cacti have modified leaves (spines), so it is unclear what is meant here. Also, Mammillaria have no noticeable photosynthetic leaves on any part of the plant, because they have been so heavily modified into spines. So, this should be clarified.

– Thank you, this was a major misphrasing. We have provided an entirely new sentence clarifying the impact of diurnal temperature variation in the two examples (lines 222-224).

Line 336, the authors mention that soils with high sand content are likely to be deserts. This is not clear at all. Sandy soils are not the foundation for deserts, as this sentence reads. Rather sandy soils may lead to edaphically dry areas, regardless of whether or not they are located in deserts. Sandy soils are global and do not necessarily correspond to deserts. Check the Guianan Shield, Amazonian white sand forests, and the Southeastern United States as examples in the Americas...

– Thank you, this is a much better way to phrase the hypothesised impact of sandy soils in cactus diversification. We have removed the reference to deserts and replaced it with “edaphically dry areas” (line 260).

The literature on Cactaceae is super extensive, so the authors should try to avoid overciting their own work. For instance, in the conclusions the authors state, “Cacti are the subject of intense macroevolutionary research, revealing a multitude of factors shaping diversification rate, some of which are correlated (Hernández-Hernández et al., 2014).” One paper is not intense macroevolutionary study...

– We have extended the citations provided here to include many more important and recent publications (lines 393-394).

Suppl. Figure 1, It would be good to provide a table to show where the rate shifts have occurred or label this with the clade name on the figure. Otherwise, the figure is hard to evaluate, other than seeing rate shifts have occurred. This is important given the authors are suggesting certain features, such as growth form, are associated with rate shifts.

– We have now included clade names in the Supplementary Figure of rate shifts (Supplementary Figure 1), which makes it much clearer to understand the lineage distribution of radiations. However, we did not include this in Figure 1 of the manuscript, as it introduces significant “noise”, as we have found out previously in other articles. It is difficult to plot rate shifts in a large phylogeny informatively, and branch colours alone can give a good indication of rate heterogeneity.

– Furthermore, we have summarised each rate shift in a table (Supplementary Table 1), including details on timing, sampled genera and species, and taxonomic group.

Suppl. Figure 2, What is C? It is not listed in the legend.

– Thanks, we have corrected this. It is the tribe Cactaeae.

Figure 1, the cholla and the *Opuntia* photos are located in the wrong clades based on their phylogeny. Please check the *Eriosyce* clade to make sure the photo used is an *Eriosyce*. It is not obvious from the figure.

– Thank you for noticing this. We have remade this figure, after also noticing that lead author Thompson placed several more photos in the wrong clades too. We have confirmed that the *Eriosyce* is indeed an *Eriosyce*. We have also removed the insets of latitude and size, given that these are no longer the top two most important factors.

Figure 5, legend, arboreal generally refers to something living in a tree (often animals). I think the more appropriate term here would be arborescent, when referring to growth form.

– Thank you, we did not notice this. We have changed it.

Given the controversy around the use of BAMM, it could be worthwhile for the authors to discuss their use of the tool.

– We acknowledge that BAMM can be problematic as an estimator of diversification rates, which we discuss in detail in lines 371-387. Furthermore, it remains one of the most-widely used methods for larger phylogenies like ours, for a number of reasons we discuss. We have added some discussion about issues with BAMM, and described potential alternatives and discussed why we did not (or could not) use them. We have also addressed how we mitigated major concerns by using a highly-conservative prior of a single rate shift (lines 371-387).

– We are confident in the utility of BAMM for analysing our phylogeny.

The Chromosome Counts Database has not been updated with any new records for Cactaceae, since the mid-2000s, so it might not be the most reliable source for counts. The authors also used the median count for each species in their analysis. Many studies have shown that multiple species are often represented by numerous ploidal levels often found within “one” species, so it may perhaps be more accurate to treat multiple counts separately (although not as easy).

– We investigated several sources to acquire chromosome counts for Cactaceae, including CCDB and the recently-published and thorough literature search by Castro et al. (2020). We selected the CCDB due to its relative coverage, because even though Castro et al. report counts for more species, the authors appear not to have done any taxonomic correction. A large fraction of their reported species were synonyms or species with very outdated names. In many cases, divergent counts were reported for synonyms under the same species. After

we attempted to clean this data, it resulted in fewer overall species, with many that were unreliable due to taxonomic issues.

– We do believe that the CCDB is a very highly reputable source of chromosome counts, which has facilitated many studies, including for Cactaceae (e.g. Las Penas et al. 2023, <https://academic.oup.com/biolinnean/advance-article/doi/10.1093/biolinnean/blad070/7242992>).

– Importantly, in our new XGBoost analysis with better-sampled GBIF data chromosome count is no longer predictive, in line with your previous comment about polyploidy. Therefore, we did not revisit this source of data.

The authors used Hunt (2016) for their taxonomy, which is now outdated. Why not use a more recent and expertly curated (from the Cactaceae community) list for the family, such as Korotkova et al. 2021?

– We do appreciate that this may be a cause for concern. But we hope you can understand why we continue to use this taxonomy.

– Firstly, we began this study a long time ago and it experienced a hiatus whilst first author Thompson was finishing his PhD and starting a new job. At the point the molecular sequences were gathered, the phylogeny was reconstructed, and the morphological data collected, Hunt (2016) provided a robust taxonomy and Korotkova et al. (2021) was not yet published.

– It would be unfeasible to re-collect the sequence and morphological data, which may not necessarily change the phylogeny significantly and may in-fact lose sample size. It would also be an impossible amount of work for lead author Thompson, who is currently juggling both a lecturing job and research job in epigenomics, and applying for long-term jobs.

– Automatic taxonomic reconciliation could potentially help this, but in our experience it can be quite inaccurate, especially with a “custom” taxonomic list (i.e. not yet available in a standard R package such as taxize).

– Finally, we do not believe that the results would change significantly with all the work to align the phylogeny and data to Korotkova et al.’s taxonomic list. The phylogeny is highly-similar to most previous hypotheses. Fine-scale changes brought on by remaking the phylogeny with the Korotkova et al. taxonomy are unlikely to change the results significantly, and it is unclear whether this would increase taxonomic breadth of sampling from Genbank.

– We really understand the concerns here, but hope you appreciate our reasons for wanting to maintain the current phylogeny.

It would be good for the authors to describe how they measured growth size in Cactaceae, especially given that they found it to be significant. What is meant by small vs. large and intermediate, as given in the results and discussion. I presume that height and length were synonymous given the description in the methods, but this is not totally clear. Are there actual values associated with these categories, or are the criteria subjective? One person's large might be another person's intermediate... This could be greatly clarified here, instead of citing other papers. This data also does not appear to have been made available on Github, so it is hard to evaluate.

– When discussing the relationship between plant size (either height or length as specified in methods), we decided to keep this broad instead of provide specific values, because QuaSSE models are described by the author (Rich FitzJohn) as being able to provide only the general trend, rather than the exact relationship (described in lines 271-273).

– Therefore we used “small” to describe cacti smaller than this inflection point, “intermediate” to describe cacti with sizes at this inflection point, and “large” above this inflection point. We deliberately kept this broad, and referred to the figure, and very quickly explained how growth form may be a better predictor. We did however explain this in the updated manuscript (lines 275-281), and provide a visualisation of the variation in the updated Supplementary Materials (Supplementary Figure 3).

Pilosocereus is given as an example from the Caribbean of a rapidly evolving clade. In general the Caribbean is not super species rich, as compared to continental high diversity areas, as the authors clearly state. That also goes for Pilosocereus, which is much more species rich in South America, for example. The tree used here only has three species of Pilosocereus from the Caribbean, so it is a little baffling how that clade is highlighted for rapid speciation in the Caribbean region. Could this be a misinterpretation of the results? Looking at the map of points, the Caribbean is quite red, but only the Pilosocereus clade is really represented in the phylogeny (Leptocereus only has a couple of taxa, Melocactus contains only one Antillean species in the tree). So these larger radiations in the Antilles are essentially not represented. Thus, it seems that the red points representing diversification in the Caribbean must be lots of individual points for just three species of Pilosocereus? This is confusing, and I don't think it reflects reality. Perhaps I don't fully understand the methodology of how the points were included here. It would be nice to see some clarification.

– Thank you, it is definitely quite difficult to plot how speciation rates vary across space, in contrast to species richness. We plotted median rates per grid cell, and in an area with relatively few species (such as the Caribbean), the colour of grid cells will be shaped strongly by the relatively-few species inhabiting them.

– The reason why Caribbean *Pilosocereus* may appear to have unexpectedly fast speciation rates, given the paucity of sampling in our tree, is because we accounted for incomplete sampling with genus-level sampling fractions. BAMM “knows” there are more species than sampled, and estimated rates accordingly. *Pilosocereus* diversification rates are known to be rapid from a recent study (Romeiro-Brito et al., 2023). However, it may be true that a mainland subset of *Pilosocereus* is diversifying more rapidly than the Caribbean representatives, and the Caribbean species are being “carried along” in the BAMM analysis.

– In an ideal BAMM analysis, the sampling fraction file may be partitioned at a finer level e.g. separating Caribbean and mainland *Pilosocereus*. But this will be impossible to implement across all genera in the tree, it would be subjective (e.g. how to split distributions into discrete regions), and incomplete geographic data would impact designation. Genus-level is the finest level of accounting for incomplete sampling that we could perform, although it is clearly not without its own issues in cases like *Pilosocereus*.

– We acknowledge that this result may be misleading, and have removed the map figure (Figure 4) from the main manuscript. It is also not particularly relevant anymore, now we have removed latitude as a predictor variable. We have instead put Figure 4 in an appendix. We already assessed whether the potentially-spurious BAMM-estimated rates for *Pilosocereus* shape results by performing the Outlier Sensitivity Model, even though XGBoost accounts for outliers, and found there are minimal impacts.

In general, the manuscript appears to be lacking in clear methodology, some misinterpretation or misrepresentation of the literature and basic morphological features of cacti (and not broad enough coverage of the literature) and results, and does not present a terribly convincing argument that the authors have found novel patterns here. In the conclusions, I am left wondering what was discovered in this manuscript, other than a latitudinal gradient corresponding to diversification rates. Across the Americas cacti generally started to diverge significantly in the Miocene and then most of the overall diversity seems to have been generated in the Pliocene and into the Pleistocene epochs, so why would only lower latitudes produce this pattern in this dataset? This is not well explored. When looking at the scatterplot for plant size in Figure 1, the vast majority of points are all in the lower size range, then with a few outliers at the higher end, so are larger taxa really diversifying faster? If I look at the red parts of the tree (faster diversification), those all correspond to small plants in the phylogeny (*Mammillaria*, *Echinocereus*, *Rhipsalis*,

Eriosyce, etc.) and which are not all at lower latitudes (i.e., Mammillaria and Echinocereus are most abundant in Mexico). So, is there truly a latitudinal pattern here? See also my comment above about potentially biased data with Pilosocereus over-representation in the Caribbean.

- Thanks, we hope that you think we have explained the issue with size (“small”, “intermediate” and “large”) more clearly in the updated manuscript.
- We have added more-detailed methods, especially regarding XGBoost.
- We also realise we over-cited certain authors, and preferentially cited newer articles instead of giving broad coverage of the literature, and have added more citations in the introduction and discussion.
- After re-running analyses with the new GBIF sampling step and without latitude, we have recovered novel and much more insightful results, instead of largely confirming previous hypotheses. This sampling step allowed us to better account for spatial biases, and has led to the emergence of important climatic variables such as “mean diurnal air temperature range”. By excluding latitude, the covariates shaping the diversification patterns were revealed.

Reviewer #3 (Remarks to the Author):

General comments

The manuscript by Thompson et al. investigates the relative importance of a range of abiotic and biotic variables in influencing the rates of speciation on the iconic botanical family Cactaceae. By applying phylogenetic comparative methods, in combination with boosted regression tree analyses, the authors describe a complex set of factors as potential drivers of speciation, particularly latitude and plant size. Overall, the manuscript is well written, and I recognize the effort that went into creating a comprehensive dataset of variables. However, I did find some relevant methodological shortcomings that, in my opinion, preclude acceptance of the manuscript as it currently stands. As you will see from my comments below, my main concern relates to the interpretation of the effect of some variables, in addition to an overall shallowness in method description and data exploration. Please, do not interpret these comments as a discouragement, but as recommendations for the improvement of the work.

Major points

My first major point relates to the use and finding of latitude as a predictor for speciation rates. Latitude is of course correlated with species richness in most taxa in such way that one of the most examined macroecological pattern is the latitudinal diversity gradient. However, as a predictor, what does latitude refers to? This point is relevant because as stated multiple times in the manuscript, latitude is one of the main “drivers” of Cactus diversification. But how can latitude “drive” diversification rates? Latitude itself does nothing that makes any biological sense, but variables correlated with latitude might do (e.g. temperature, humidity, energy, seasonality). Beyond the potential effect related to the time-for-speciation hypothesis (i.e. the tropics has been geologically more stable through time, allowing more time for species to accumulate), I see no other way that latitude could be a “driver” of evolutionary rates. In fact, the results clearly point to that, when the authors claim that: “...we find latitude is the strongest predictor in both XGBoost models, suggesting a primary role of latitude over covariates...after accounting for interactions in the complex model the relative importance of several climatic covariates is reduced” (lines 259-253). This is perhaps an indication that latitude is explaining more variability because it correlates with several other climatic covariates, and therefore, has a much better power for separating residuals in the boosted regression tree analysis.

Having said that, I wonder what happens to your results if you remove latitude and only keep the climatic variables in the model?

- We thank you for such a comprehensive and fair commentary on our MS.
- After much deliberation, we have decided to remove latitude as a predictor and rerun all analyses. We agree that “latitude” is not a driver in itself, and must be driven by covariates.
- This involved major revision of the MS from all-but the earliest steps, and has made the MS immeasurably stronger. By removing latitude, we reveal the impact of many covariates such as diurnal temperature range, bringing a much more nuanced discussion of cactus evolution.

I would also suggest looking at a covariance structure between explanatory variables, to check if there is a high degree of collinearity between them (maybe through the variance inflation factor - VIF). This would potentially help to disentangle what variables correlated with latitude are more likely influencing speciation.

– Thank you for this advice. While we did explore looking into covariance structures, we read much more deeply into boosted regression trees, and are confident that this approach can extricate predictors accurately in the face of covariance, including in complex ecological datasets (Elith et al., 2008 *J Anim Ecol.* <https://doi.org/10.1111/j.1365-2656.2008.01390.x>).

– This ability means we can maximise the number of sampled variables we include despite covariance, and can confidently analyse them with boosted regression trees.

This point is also relevant because of the large number of variables included in the model. The authors claim that 11 variables were identified to “significantly contribute to the diversification of Cactaceae” (lines 22-24). However, upon closer inspection (Fig. 2), most of these variables have a relative importance of less than 5% in the final model. This only represents 1.1% of total variance explained, considering that their best model only explains a total of 22% of the variability in rates. This is quite a low percentage for many of the factors that have been discussed as potential “drivers” of diversification. I agree with the authors that modelling evolutionary rates is complex, and their study is a testament of that. However, I believe that focusing the discussion on factors that were identified as “above chance expectation” might not be the best strategy in this case. This makes the discussion unnecessary long and complex, leaving the impression that almost anything could be contributing to higher diversification in the group, which might not necessarily be the case given the low explanatory power. Hence, my suggestion here is to try and narrow down the number of variables analyzed (maybe with the suggestion above) before including them in the final boosted regression tree model. This will hopefully provide more resolution and a better final model.

– Thank you, we appreciate this deep examination of our results and the abilities of XGBoost. We agree that this MS is extremely wordy as a result of the discussion section investigating 11 variables, many of which are of minor effect.

– We followed Siqueira et al. (2020, <https://www.nature.com/articles/s41467-020-16498-w>) in distinguishing variables significantly predicting tip rates originally. But they went on to investigate and discuss only one major variable, whereas we chose to discuss 11 for completeness.

– In this revised MS, the list of significant variables we discussed was smaller, on account of differently-processed GBIF data and the removal of latitude. Additionally, we reduced this to five variables, by determining significance differently. Now, we ask whether the upper and lower quartile gain estimates from the 1,000 XBoost bootstraps do not overlap the “above chance expectation” threshold. I.e. are all of the lower quartile, mean and upper quartile gains above $1/\text{number of variables}$ (lines 530-533).

– This was a very good recommendation, especially in conjunction with the advice from reviewer 3 about removing latitude from the pool of variables, and has made the MS much stronger.

Now, back to the latitude topic, by looking at Fig. 4, it seems like the trend of higher tip-speciation is found in the Caribbean for +25° and in regions of potential endemism in South America (Brazilian Cerrado?) for -25°. Have the authors thought about including endemism and species richness, for example, as potential explanatory variables for speciation rates? This would potentially give more context to “latitude” with more relevant ecological predictors. In fact, I cannot find anywhere in the manuscript a list of all the 39 variables, so maybe the authors should consider including a supplementary table with the variables and their hypothesized effect. This could also help to narrow down the number of variables included in the model, after critically evaluating the role of each one.

– We have removed latitude as a predictor entirely, but thank you for these points to think about. We have also provided a list of predictor variables and their hypothesised effects in Supplementary Table 2.

Regarding the relationship with size, although the authors find that “An inverted modal relationship between Cactus size and speciation rate is best supported, with fastest rates in the smallest and largest Cacti, and reduced rates in intermediate sizes (Figure 3)”, this does not seem to agree with the results for tip-speciation (Fig. 1). The relationship of tip-speciation with size seems to suggest that only smaller species tend to have really high speciation, which is interesting but completely ignored in the manuscript. This is particularly important because the authors discuss “larger forms speciating rapidly, contrary to much research on size evolution across the Tree of Life.” (lines 473-474). In my opinion, this point should be further explored because the tip-rates suggest that your patterns are not contradicting expectations, as suggested.

– The relationship between size and diversification rate in cacti is complex. Size variation is strongly shaped by growth form (see Supplementary Figure 3), and previous research has strongly linked larger growth forms (arborescents, shrubs and columnars) with elevated diversification rates (Hernandez-Hernandez et al., 2014).

– In the original MS, we replicated this result using BiSSE and much wider sampling than Hernandez-Hernandez et al. (2014), although HiSSE confirmed the role of hidden states in shaping this relationship.

– An important factor to consider is that in our original MS, the tip-rates plot was misleading, because it does not control for shared ancestry. Thus, any patterns derived from this are, at

face value, incorrect. The SSE models we employed do account for shared ancestry estimated from the phylogeny.

– QuaSSE is a rigorous and statistical method of assessing relationships between plant size and diversification rates, while controlling for the impacts of shared ancestry. While the figure certainly suggested there is a negative relationship between plant size and diversification rates, this did not control for shared ancestry.

– However, QuaSSE is limited to providing model comparisons and general trends, instead of exact relationships (FitzJohn 2010). The modal-model QuaSSE provides is intuitive given that growth form shapes the relationship between plant size and diversification rates, given that it is clearly shaped by smaller forms vs larger forms.

– But, from our original MS and that of Hernandez-Hernandez et al. (2014), we would expect a faster diversification rate in cacti with larger growth forms than smaller, which QuaSSE cannot model. Instead, we see this “negative modal model” where rates are slowest in medium sizes. This is why in the original MS, we discussed how continuous size was a better predictor, but limited the discussion of the nature of this relationship to growth form, a weaker predictor but one which cannot be ignored when investigating cactus size.

– But in the new MS with better-sampled spatial variables, lack of latitude, and our more-stringent delimitation of which predictors are significant, growth form is no longer significant and discussion around this has been removed.

Finally, I was also worried about the overall lack of support for the phylogenetic tree used in the manuscript. The authors suggest that the phylogeny was “moderately well-supported”, but the reality is that the majority of nodes (almost 70%) had very low support values.

– Thank you, we realised we effectively ignored the impacts of phylogenetic uncertainty in our analyses. This was also raised by reviewer 1, and we have addressed it explicitly in lines 360-370, and it has made our MS much more transparent and stronger as a result.

In addition, the time-calibration step only used two points of calibration, which sounds like very little information for such large phylogeny. All of this is incredibly relevant, given that all downstream analyses are dependent on this phylogeny, which has the potential for error propagation if the support is so low and the calibration is uncertain.

– Divergence estimation in Cactaceae, and plants generally - especially succulent plants - is very difficult. Cacti lack fossils, and divergence estimations for crown ages of cacti have varied dramatically as a result, from 28.6-53Mya (Lavor et al., 2018 *J. Biogeogr.* <https://doi.org/10.1111/jbi.13481>; Yao et al., 2019 *Mol. Phylogenetics Evol.*

<https://doi.org/10.1016/j.ympev.2018.12.023>). Secondary calibrations are typically relied upon, but these rely on accurate molecular clock analyses with widely-sampled fossils and taxa. Only recently has a reconstruction been published that investigated every angiosperm family (thus optimising the molecular clock across angiosperms), with a wide and curated sample of fossils, and a relaxed molecular clock in a Bayesian framework (Ramirez-Barahona et al., 2020 *Nat Eco Evo* <https://doi.org/10.1038/s41559-020-1241-3>).

– We used these few calibration points for our divergence estimation, because they are considered highly robust and accurate. We recognise that internal nodes are not calibrated, but timings of these are constantly in flux as sampling and models improve, and Ramirez-Barahona et al. (2020) did not sample nodes within Cactaceae, due to computational limits.

– However, regardless of which absolute ages were calibrated, our relative results (i.e. relative diversification rates) could be replicated by constraining the crown with an arbitrary age (e.g. of 1Ma). We were careful not to over-interpret temporal variation in diversification rates, as two of the authors (Thompson and Priest) have successfully done before in orchids (<https://www.pnas.org/doi/10.1073/pnas.2102408120>), which have a robust time-frame for internal nodes as well as their crown and stem.

– All of our analyses in this MS can be considered agnostic to absolute divergences. There would be no benefit to constraining internal nodes according to previous estimates (e.g. Arakaki et al., 2011), because these relied on the molecular clock variation to estimate their ages, as we did here. It may even introduce biases of their taxonomic/molecular sampling into our study.

Other Points

Overall, I felt like the methods section could be improved. For example, there is only one sentence explaining how you accounted for interactions in the boosted regression tree analysis, despite this being an important aspect of the manuscript. I suggest being more specific in the methods to allow for clarity and reproducibility.

– Thank you, we recognise that this was not adequately explained for such an important point. We have added explanations in the introduction (lines 101-103) and edited the methods, including by improving the subheading name (line 494) and by providing a better explanation in the methods (lines 526-533). We believe that this adequately explains how complexity was compared.

Line 151: I failed to find your sensitivity analyses excluding outliers in the supplementary data. Please double check.

– This was not clearly described originally, but it is present in the folder “XGBoost Relative Importance Tables”.

Line 1149: “best-fitting” instead of “best-fitting”.

– Thanks, we have corrected this.

Figure 5: Given that HiSSE had a better support, shouldn't this plot show the HiSSE results instead? This also applies for the results section. If HiSSE had a better support, it means that unobserved traits are more influential to diversification rates than these two, which is also indicated by the XGBoost results.

– We originally did not present the HiSSE models, because our XGBoost analyses better-modeled the complexity and multifactoriality of forces shaping cactus diversification. HiSSE models simply confirmed this, but the exact parameter estimates added little to the debate. They also did not plot any uncertainty of state-dependent diversification rates, which BiSSE MCMC models did. We presented BiSSE MCMC plots to aid visualisation of the relationship, with XGBoost and HiSSE described purely to illustrate the complexity.

– However, since re-running the analyses, growth form and polyploidy are no longer significant and all BiSSE and HiSSE models have been removed.

I hope this review was helpful.

– This review was extremely helpful, not just for this manuscript but for several others we have in the pipeline.

– We are very grateful, thank you.

REVIEWER COMMENTS

Reviewer #1 (Remarks to the Author):

The authors have substantially improved this manuscript, particularly in reanalyzing predictions using machine learning and subsequent textual alterations. This has notably addressed the spatial sampling bias of environmental variables, resulting in new findings that are more robust and understandable than the previous version.

Furthermore, I commend the authors for their thorough and satisfactory responses to my comments and concerns, with one exception highlighted in comment #13. In this regard, the author's response is unsatisfactory, as a theoretical problem that must be addressed persists.

1 – LI. 297-300 - My concern lies with the statement which refers to the potential role of greater pollinator mobility in promoting reproductive isolation. After reviewing the cited reference, I could not find substantiating evidence for this claim. Instead, Xu et al. (2012) propose that pollinator shifts, facilitated by changes in floral characteristics, may contribute to pollinator-driven speciation in plants by establishing strong pre-zygotic isolation mechanisms. Therefore, I kindly request the authors to revisit this statement and either provide further clarification or revise it accordingly to align with the findings of Xu et al. (2012).

It is worth noting that although the reference to Xu et al. (2012) was not included in the bibliography, I could identify it in a prior publication by one of the present study's authors.

Reviewer #3 (Remarks to the Author):

As Reviewer #3 in the first round, I am pleased to see that the authors have addressed most of my initial concerns. I have carefully reviewed their responses to all reviewers. In general, they provided detailed comments and made relevant changes that helped improve the work. However, I still have some points that require attention by the authors. I am happy to recommend this work for publication after these comments have been addressed.

My main point is still related to the Boosted Regression Tree results. I appreciate the authors' response to my suggestion to remove latitude as a predictor. This confirmed my initial suspicion that latitude was contributing significantly to the variation due to its composite nature. Indeed, the fit of the main model remains relatively similar ($R^2 \sim 0.21$) compared to when latitude was included. I also agree with the decision of choosing "significant" factors as those that do not overlap with the bootstrapped confidence intervals, which made the manuscript more streamlined. However, my concern about the low explanatory power of the variables remains, and I believe this should be included in the caveats section. For example, the most important variable (Diurnal air temperature range) still explains only ~2% of the total variance in predicting speciation rates. In my opinion, this underscores the difficulty of predicting diversification rates and highlights the significant unexplained variance in cactus diversification. Therefore, I suggest that the authors focus their conclusions on the complexity of diversification regimes rather than solely on the detected "drivers". This is just a simple change in the framing of the manuscript, which I think will strengthen it and make it more transparent if the authors agree.

In response to one of my previous comments, the authors also claim that patterns derived from the tip-rates are, at face value, incorrect because they do not control for shared ancestry. However, if that is the case, the Boosted Regression Tree analysis could be misleading. Careful not to shoot yourself in the foot. I disagree that the tip-rates are incorrect; rather, they seem to display a different pattern compared to SSE models. Therefore, my suggestion is to also examine the relationship predicted by the Boosted Regression Tree analysis. If QuaSSE is limited to providing general trends, your XGBoost analysis should return the actual relationships found for tip-rates using the main predictors. I believe

it would be a good idea to include the partial dependence plots for the main variables even if it is just as supplemental material.

Finally, I believe that my suggestion to include endemism and species richness as potential explanatory variables was not satisfactorily addressed. I am not necessarily saying that these variables should be included, but it would be at least interesting to hear what the authors think about these variables and why they should be included or not.

Lines 520-521: "decision tree" is repeated.

Reviewer #1:

The authors have substantially improved this manuscript, particularly in reanalyzing predictions using machine learning and subsequent textual alterations. This has notably addressed the spatial sampling bias of environmental variables, resulting in new findings that are more robust and understandable than the previous version.

- Thank you, we agree that the manuscript is much stronger as a result of these edits and appreciate the helpful suggestions.

Furthermore, I commend the authors for their thorough and satisfactory responses to my comments and concerns, with one exception highlighted in comment #13. In this regard, the author's response is unsatisfactory, as a theoretical problem that must be addressed persists.

- Thank you, we enjoyed making these changes and replying to the comments, they made the manuscript a lot stronger.
- We apologise for this remaining error and have corrected it (see below).

1 – LI. 297-300 - My concern lies with the statement which refers to the potential role of greater pollinator mobility in promoting reproductive isolation. After reviewing the cited reference, I could not find substantiating evidence for this claim. Instead, Xu et al. (2012) propose that pollinator shifts, facilitated by changes in floral characteristics, may contribute to pollinator-driven speciation in plants by establishing strong pre-zygotic isolation mechanisms. Therefore, I kindly request the authors to revisit this statement and either provide further clarification or revise it accordingly to align with the findings of Xu et al. (2012).

- We agree we misrepresented the Xu et al reference, and have removed it. Given that we had correctly integrated Xu et al's findings earlier (lines 304-305), we corrected the sentence with the second reference to Xu et al, and replaced the reference with appropriate publications (lines 308-310).

It is worth noting that although the reference to Xu et al. (2012) was not included in the bibliography, I could identify it in a prior publication by one of the present study's authors.

- We also apologise for not putting this reference in the references section. We have checked the remaining references and found this was the only one missing.

Reviewer #3:

As Reviewer #3 in the first round, I am pleased to see that the authors have addressed most of my initial concerns. I have carefully reviewed their responses to all reviewers. In general, they provided detailed comments and made relevant changes that helped improve the work. However, I still have some points that require attention by the authors. I am happy to recommend this work for publication after these comments have been addressed. My main point is still related to the Boosted Regression Tree results. I appreciate the authors' response to my suggestion to remove latitude as a predictor. This confirmed my initial suspicion that latitude was contributing significantly to the variation due to its composite nature. Indeed, the fit

of the main model remains relatively similar ($R^2 \sim 0.21$) compared to when latitude was included. I also agree with the decision of choosing “significant” factors as those that do not overlap with the bootstrapped confidence intervals, which made the manuscript more streamlined.

- Thank you for your careful and thorough consideration of our reply. We really appreciate it and agree that the work is much stronger now as a result, especially after removing latitude as a predictor, and streamlining the large number of significant variables by applying a more stringent definition of significance.

However, my concern about the low explanatory power of the variables remains, and I believe this should be included in the caveats section. For example, the most important variable (Diurnal air temperature range) still explains only ~2% of the total variance in predicting speciation rates. In my opinion, this underscores the difficulty of predicting diversification rates and highlights the significant unexplained variance in cactus diversification. Therefore, I suggest that the authors focus their conclusions on the complexity of diversification regimes rather than solely on the detected “drivers”. This is just a simple change in the framing of the manuscript, which I think will strengthen it and make it more transparent if the authors agree.

- We thoroughly agree with the concerns that the total variance explained is quite low, and we missed an opportunity to use this as evidence of the complexity of macroevolution, instead focussing on “what we now know”. We slightly missed the point of our own manuscript, that macroevolution is complicated.
- We have now explicitly described the low variance explained, and we have added details to shift the narrative to reflect this (lines 201-203, 217-222, 396-406).
- We agree that the complexity aspect is the overarching theme of these results, however, we have decided to keep the discussion of individual drivers, albeit with some narrative reshaping in light of your helpful comments.
- To do this, we have made changes to present these five variables as hypotheses for further research, instead of “confirmed” drivers (lines 221-222). We do believe that our analyses have done a reasonable job of cutting through some of the complexity to identify these drivers, and our integration of wider literature will serve as a good basis for future investigation. However, further research is definitely needed to explore the nuances of these relationships, and confirm the importance of any of them.
- Author JBT recently won funding from the British Cactus and Succulent Society to explore one of these drivers in significantly more depth, and his student has benefitted from the references and theory in the hypotheses we have made here.
- Reframing the narrative to encourage further exploration, instead of taking these results as “confirmed”, is especially important given the limitations of QuaSSE and the differences with non-phylogenetically-controlled, XGBoost-predicted, relationships. I will discuss this aspect in more detail later on in this reply.
- I believe this aspect of the manuscript has benefited greatly from your comments, and is a more honest representation of the complexity we encountered.

In response to one of my previous comments, the authors also claim that patterns derived from the tip-rates are, at face value, incorrect because they do not control for shared ancestry.

However, if that is the case, the Boosted Regression Tree analysis could be misleading. Careful not to shoot yourself in the foot. I disagree that the tip-rates are incorrect; rather, they seem to display a different pattern compared to SSE models. Therefore, my suggestion is to also examine the relationship predicted by the Boosted Regression Tree analysis. If QuaSSE is limited to providing general trends, your XGBoost analysis should return the actual relationships found for tip-rates using the main predictors. I believe it would be a good idea to include the partial dependence plots for the main variables even if it is just as supplemental material.

- We have now provided the predicted speciation rates across the variation of different predictors. We appreciate that it is relevant and powerful, despite not accounting for shared ancestry.
- We have decided to provide this in the updated supplementary materials and focus on the QuaSSE results, because they control for shared ancestry and are thus more appropriate for macroevolutionary analyses. However, we do not provide these just as an afterthought, we explicitly describe how they differ from QuaSSE estimates (lines 188-190 and 217-222).

Finally, I believe that my suggestion to include endemism and species richness as potential explanatory variables was not satisfactorily addressed. I am not necessarily saying that these variables should be included, but it would be at least interesting to hear what the authors think about these variables and why they should be included or not.

- We agree that we did not adequately address the comments about endemism and species richness as potential drivers, and apologise for this.
- We appreciate that the richness of a region, and endemism of a species, may drive speciation, perhaps through competition, niche differentiation, geographic isolation, and genetic effects, among others.
- We believe we have captured the evolutionarily-important aspects of endemism already by estimating geographic range size. While we appreciate that a species can be endemic with a larger range size, e.g. endemic to a large mountain range, we believe that geographic range size is the relevant evolutionary component of endemism when it comes to macroevolutionary dynamics (a good discussion of this is available here <https://doi.org/10.1038/s41467-023-41225-6>).
- Furthermore, we believe our estimates of geographic range size are robust and objective. We estimated geographic range size using the AOO method, defining by the number of grid cells inhabited, accounting for some of the information lost by incomplete GBIF sampling. By scrutinising the GBIF data in the cleaning stage and comparing with independent descriptions of distributions, we are confident that many of the species with small estimated ranges from GBIF data do indeed have small ranges, and are not just poorly sampled.
- XGBoost and QuaSSE require a measure at species level, which would be difficult to capture with available data for richness. We explored several ways to do this, e.g. estimating the number of species within a radius around a record, then averaging for each species. If GBIF data were perfect (i.e. every single cactus had been mapped),

this could be a robust, efficient and novel way to capture this information as a continuous variable.

- However, the taxonomic and collection gaps in GBIF make it difficult to do this effectively, and we also struggled to objectively justify a radius. I.e. should a conspecific species influence a species-level estimate of richness if it is beyond 1km from the focal species, or 5km, 10km, or longer if “ecologically connected” despite distance (e.g. by pollinators)? And should we consider the richness of species outside Cactaceae, which may also confer ecological pressures onto the cacti? We really struggled to quantify a species-level measure of richness.

Lines 520-521: “decision tree” is repeated.

- Thank you for spotting this, we have corrected it.

Returning to your comments from the previous submission, where you said:

I would also suggest looking at a covariance structure between explanatory variables, to check if there is a high degree of collinearity between them (maybe through the variance inflation factor - VIF). This would potentially help to disentangle what variables correlated with latitude are more likely influencing speciation.

And we replied with:

– Thank you for this advice. While we did explore looking into covariance structures, we read much more deeply into boosted regression trees, and are confident that this approach can extricate predictors accurately in the face of covariance, including in complex ecological datasets (Elith et al., 2008 J Anim Ecol. <https://doi.org/10.1111/j.1365-2656.2008.01390.x>).

– This ability means we can maximise the number of sampled variables we include despite covariance, and can confidently analyse them with boosted regression trees.

- While we are still confident in XGBoost, and our decision to maximise data inclusion to model complexity, we did confirm that there is little evidence for correlation among variables in the data. Crucially, variables identified as significant are not correlated.
- We have provided correlation matrix plots to support this in the Supplementary Materials, and described it in the text (lines 156-158, 547-550).

Reviewers' Comments:

Reviewer #3:

Remarks to the Author:

Once again, I appreciate the Authors' responses and consideration to my comments on the previous round. I only have a few minor points to suggest.

- Line 119: Figure 1 is not showing node support, so maybe remove the reference to this figure here or include a figure with node support in the supplemental.
- Be more specific about where to find your outlier sensitivity model (as per previous comment) in the main text so readers don't need to dig to find it.
- Line 180: it is not the biodiversity of cacti that is generated by moderate climatic variability, it is speciation rates that potentially get accelerated. These things have different connotations, and it is important to be clear about your results that only refer to macroevolutionary rates.
- Please provide more details about your State-dependent diversification methods section. For example, how were models compared? Did you run one model per trait and compared each one with a null model? These details are not very clear in the methods description.
- Double check the supplementary material for accuracy (e.g. mention to table xx in the legend of Supplementary Figure 1).

We thank reviewer three for taking the time to do a thorough reading of our manuscript and providing great comments for improvement.

Reviewer #3 (Remarks to the Author):

Once again, I appreciate the Authors' responses and consideration to my comments on the previous round. I only have a few minor points to suggest.

- We appreciate the comment.

- Line 119: Figure 1 is not showing node support, so maybe remove the reference to this figure here or include a figure with node support in the supplemental.

- We have removed reference to the figure here.

- Be more specific about where to find your outlier sensitivity model (as per previous comment) in the main text so readers don't need to dig to find it.

- We have provided the specific location for this supplementary file.

- Line 180: it is not the biodiversity of cacti that is generated by moderate climatic variability, it is speciation rates that potentially get accelerated. These things have different connotations, and it is important to be clear about your results that only refer to macroevolutionary rates.

- We have corrected this to describe speciation rate accelerations instead of the previous statement.

- Please provide more details about your State-dependent diversification methods section. For example, how were models compared? Did you run one model per trait and compared each one with a null model? These details are not very clear in the methods description.

- We have edited this to make it clear we are referring to seven models applied to each of the five traits.

- Double check the supplementary material for accuracy (e.g. mention to table xx in the legend of Supplementary Figure 1).

- We have checked the supplementary material, which resulted in swapping the order of two figures to make it more logical and corrected the "table xx" placeholder.